# Timescales of outlet-glacier flow with negligible basal friction: Theory, observations and modeling

Johannes Feldmann[1] and Anders Levermann[1,2,3]

[1]Potsdam Institute for Climate Impact Research (PIK), Potsdam, Germany
[2]Institute of Physics, University of Potsdam, Potsdam, Germany
[3]LDEO, Columbia University, New York, USA

**Correspondence:** Johannes Feldmann (johannes.feldmann@pik-potsdam) & Anders Levermann (anders.levermann@pik-potsdam.de)

**Abstract.** The timescales of the flow and retreat of Greenland's and Antarctica's outlet glaciers and their potential instabilities are arguably the largest uncertainty in future sea-level projections. Here we derive a scaling relation that allows the comparison of the timescales of observed complex ice flow fields with geometric similarity. The scaling relation is derived under the assumption of fast, laterally confined, geometrically similar outlet-glacier flow over a slippery bed, i.e., with negligible basal
friction. According to the relation, the time scaling of the outlet flow is determined by the product of the inverse of 1) the fourth power of the width-to-length ratio of its confinement, 2) the third power of the confinement depth and 3) the temperature-dependent ice softness. For the outflow at the grounding line of streams with negligible basal friction this means that the volume flux is proportional to the ice softness and the bed depth, but goes with the fourth power of the gradient of the bed and with the fifth power of the width of the stream. We show that the theoretically derived scaling relation is supported by the
observed velocity scaling of outlet glaciers across Greenland as well as by idealized numerical simulations of marine ice-sheet instabilities (MISIs) as found in Antarctica. Assuming that changes in the ice-flow velocity due to ice-dynamic imbalance are proportional to the equilibrium velocity, we combine the scaling relation with a statistical analysis of the topography of 13 MISI-prone Antarctic outlets. Under these assumptions the timescales in response to a potential destabilization are fastest for Thwaites Glacier in West Antarctica and Mellor, Ninnis and Cook Glaciers in East Antarctica; between 16 and 67 times faster
than for Pine Island Glacier. While the applicability of our results is limited by several strong assumptions, the utilization and potential further development of the presented scaling approach may help to constrain time-scale estimates of outlet glacier-flow, augmenting the commonly exploited and comparatively computationally expensive approach of numerical modeling.

## 1 Introduction

Global sea-level rise is increasingly driven by the contributions of the ice sheets of Greenland and Antarctica. Both ice sheets
are observed to be out of balance (The IMBIE team, 2018; Rignot et al., 2019; Mouginot et al., 2019; The IMBIE Team, 2020; Diener et al., 2021). This imbalance is mainly due to the increased discharge of ice into the ocean which accounts for more than half of the cumulative net mass loss from the Greenland Ice Sheet and virtually all of the loss from the Antarctic Ice Sheet. Their discharge is projected to raise global sea level by several decimeters within the 21[st] century (Bamber et al.,

2019; Levermann et al., 2020; Pattyn and Morlighem, 2020; Edwards et al., 2021; DeConto et al., 2021). Fed by snowfall,
the ice sheets spread under their own weight towards the ocean. The flow of the ice accelerates from the interior to the coast,
where the ice sheets are drained by numerous outlet glaciers that often stream through topographic confinements, e.g., valley-
shaped bed troughs, before they discharge into the ocean (Morlighem et al., 2017; Maier et al., 2019; Morlighem et al., 2020).
Such confinements tend to induce stabilizing ice-internal stresses that are transmitted upstream, exerting a buttressing effect on
the hinterland (MacAyeal, 1989; Dupont and Alley, 2005; Schoof, 2007; Gudmundsson et al., 2012; Haseloff and Sergienko,
2018; Pegler, 2018), regulating the flow speed and thus the rate of ice discharge into the ocean. Note that we refer to the
term "buttressing" as a backforce induced by laterally confined ice flow that is either floating (often referred to as "ice-shelf
buttressing") or grounded.

In both ice sheets, the ice discharge mainly takes place through fast flowing outlet glaciers like Jakobshavn, Helheim or
Kangerlussuaq Glaciers in Greenland (Catania et al., 2020; The IMBIE Team, 2020) and Pine Island or Thwaites Glaciers
in West Antarctica (Mouginot et al., 2014; The IMBIE team, 2018; Diener et al., 2021). In Antarctica large parts of the ice
are grounded below sea level with a bed topography that is sloping down in the inland direction (retrograde slope; Pattyn
and Morlighem, 2020; Morlighem et al., 2020). These two pre-conditions make the ice sheet prone to the so-called marine
ice sheet instability (MISI), a positive feedback between ice discharge and ice-sheet retreat down the retrograde bed slope
(Weertman, 1974; Schoof, 2007; Pattyn and Morlighem, 2020). The buttressing effect of laterally confined ice flow can slow
down or even prevent MISI-type destabilization (Gudmundsson et al., 2012; Haseloff and Sergienko, 2018; Pegler, 2018). The
development of a MISI was hypothesized for the West Antarctic Ice Sheet more than forty years ago (Hughes, 1973; Mercer,
1978) and meanwhile could indeed have been initiated as suggested by observations and ice-dynamic modeling (Favier et al.,
2014; Joughin et al., 2014; Seroussi et al., 2017; The IMBIE Team, 2020; Diener et al., 2021).

Determining the quantitative influence of buttressing on the timescales of confined ice flow and the related discharge in
general as well as on MISI timescales in particular requires detailed numerical computations (Favier et al., 2014; Joughin
et al., 2014; Pattyn and Morlighem, 2020). By comparing similar ice dynamics it is however possible to derive scaling relations
that comprise the complex stress and velocity field without the need of computationally expensive simulations (Feldmann and
Levermann, 2016; Levermann and Feldmann, 2019). Such relations provide a link between the characteristic scales of a glacier,
e.g., between its ice-flow timescale and its geometric dimensions. While scaling relations are valid for each glacier separately
they also allow the comparison of similar glaciers, e.g., the comparison between the timescales of two glaciers based on their
individual geometric scales.

To derive the scaling relation presented here, we conduct a scaling analysis of the governing dynamic equations of fast,
shallow and laterally confined ice flow under the assumption of small basal friction. To this end, we employ the concept
of similitude (Rayleigh, 1915), which is a common and simple procedure in hydrodynamics and engineering to investigate
the scaling behavior of physical systems (Szücs, 1980). A famous example for its application is the Reynolds number for the
Navier-Stokes equation (Reynolds, 1883; Kundu et al., 2012). A fixed Reynolds number provides a scaling relation that ensures

similar flow patterns of a fluid under variation of its characteristic properties. The scaling relation derived in the present study links the timescale of the flow of an outlet glacier to its geometry, i.e. to the spatial dimensions of its lateral confinement, and to the temperature of the ice. Earlier work that dealt with the topic of confined outlet glacier flow includes analyses of the dependency of ice flux and grounding-line dynamics on the confinement width and in-depth investigations of the role of lateral shear margins (e.g., Nye, 1965; Raymond, 1996; Joughin et al., 2004; Gudmundsson et al., 2012; Pegler et al., 2013; Haseloff et al., 2015; Schoof et al., 2017; Pegler, 2018). Laterally unconfined ice flow and its response time to small perturbations have been examined for a regime of low basal shear and driving stress in Sergienko and Wingham (2019).

In a previous study we applied the similitude principle to the one-dimensional flow problem in combination with observational data, to compare the timescales of geometrically similar, MISI-prone Antarctic outlet glaciers (Levermann and Feldmann, 2019). This one-dimensional approach, however, neglected the important buttressing effect of the outlet's lateral confinement. Our present analysis respects the buttressing effect of the grounded ice portion through the width of the confinement as one of the dominant parameters determining the timescale of the ice dynamics. The derivation of the associated scaling relation is outlined in Sect. 2. It is then tested in Sect. 3, examining the scaling of 1) observed Greenland outlet-glacier flow speeds and 2) simulated Antarctic-type MISI retreat rates. In Sect. 4 we apply the scaling relation to a set of MISI-prone Antarctic outlet glaciers to compare their potential retreat timescales, making the assumption that the retreat speed of an outlet glacier is proportional to its flow speed. The method we use to obtain the outlet-specific timescales is principally similar to but more sophisticated than the one used in Levermann and Feldmann (2019) as we here extend it to both horizontal dimensions and carry out a more profound statistical analysis of the obtained outlet-specific scales. The limitations of our approach are discussed in Sect. 5 where we also draw our conclusions.

## 2 Derivation of the scaling relation

Our derivation of the scaling relation (detailed in Appendix A1) is based on the shallow-shelf approximation of the Stokes stress balance of the ice (SSA; MacAyeal, 1989; Bueler and Brown, 2009). The SSA captures the dynamics of a fast and shallow ice flow and neglects vertical shearing inside the ice. It describes a flow which is controlled by the balance between horizontal ice-internal stresses, basal shear stresses between ice and bed, and the driving stress due to the surface gradient of the ice. Many of the outlet glaciers around Greenland and Antarctica are ice streams (Joughin et al., 2018; Mouginot et al., 2019) that slide over a slippery bed (Morlighem et al., 2013; Maier et al., 2019). While the fast ice streaming can generally produce high basal shear stresses, it has been shown that in the well lubricated, marine grounding-line regions the outlets are typically characterized by comparatively low basal shear (Joughin et al., 2004; Sergienko and Hindmarsh, 2013; Sergienko et al., 2014). The lateral confinements of the ice streams can induce high lateral ice-internal shear stresses with a strong influence on the ice dynamics, especially if the ice flows through deep and narrow bed troughs (Dupont and Alley, 2005; Gudmundsson et al., 2012; Leguy, 2015). Here we consider the basal stresses in the grounding-line region to be small compared to such ice-internal stresses and thus neglect them in the stress balance.

Typically, the ice streams in Greenland and Antarctica are much longer than wide, thus exhibiting a small width-to-length ratio $R = W/L$, as the characteristic along-flow length scale $L$ is typically observed to be a multiple of the width scale $W$ (Morlighem et al., 2017; Mouginot et al., 2019; Morlighem et al., 2020). Without loss of generality we place the two horizontal axes of the coordinate system of our analysis into the main flow direction ($x$) and perpendicular ($y$) to it (Fig. 1). Using this information in a simple dimensional analysis of the SSA, in which all quantities have been non-dimensionalized (Appendix A1), the stress balance reduces to

$$\lambda \partial_y [H(\partial_y v_x)^{\frac{1}{n}}] = H \partial_x h, \tag{1}$$

with

$$\lambda = [(2AT)^{\frac{1}{n}} \rho g D R^{\frac{1}{n}+1}]^{-1}, \tag{2}$$

where $v_x$ is the ice speed in $x$ direction and thereby along the bed trough, $H$ is the ice thickness, $h$ represents the surface elevation of the ice, $n$ is Glen's flow law exponent determining the ice rheology (Glen, 1955), and $\partial_x$ and $\partial_y$ denote the spatial derivatives in the $x$ and $y$ directions. Notably, Eq. (1) includes a single dimensionless coefficient $\lambda$, in which $A$ is the ice softness, $D$ and $T$ represent the characteristic vertical length scale and the timescale of the system, respectively, $\rho$ denotes the ice density and $g$ is the gravitational acceleration. The coefficient $\lambda$ represent the balance between 1) the leading-order ice-internal stress, i.e., the horizontal shearing of the ice flow in across-trough direction and 2) the driving stress of the ice, which is determined by the surface gradient along the trough. As long as $\lambda$ remains constant, the ratio of these two stresses will remain the same, while the characteristic scales of the system are allowed to vary. In other words, the ice dynamics of the considered system will remain similar under a scaling of the system that leaves the system's governing dimensionless number $\lambda$ unchanged (concept of similitude). Considering a reference system and applying a scaling to it (denoted by a dash), the requirement of $\lambda' = \lambda$ then yields a time-scaling relation that assures the similitude of the two systems:

$$\tau = \alpha^{-1} \beta^{-n} \left(\frac{\omega}{\delta}\right)^{-(n+1)}. \tag{3}$$

Here $\tau = T'/T$ is the ratio of the timescales, $\alpha = A'/A$ denotes the scaling of the ice softness, and $\beta = D'/D, \omega = W'/W$ and $\delta = L'/L$ represent the scaling of the confinement depth, width and length, respectively (Fig. 1, Table 1). Laboratory studies yield a range of possible values for $n$ ranging from 2 to 4 (e.g., Duval, 2013). Using the most commonly applied value of $n = 3$ we obtain

$$\tau = \alpha^{-1} \beta^{-3} \gamma^{-4}, \tag{4}$$

where $\gamma = \omega/\delta = R'/R$ expresses the scaling of the horizontal aspect ratio (width-to-length ratio) of the confinement. The timescale is thus a strongly non-linear function of the spatial dimensions of the confinement, i.e., decreasing with the fourth power of its width-to-length ratio and with the third power of the bed depth. Furthermore, the timescale is inversely related to the ice softness which means that higher ice temperatures lead to a faster timescale. This is due to the fact that a higher softness enhances the deformation rate of the ice (Glen, 1955), reducing the restraint due to lateral shear and promoting along-flow

extension of the ice, both inducing faster ice flow and discharge. The different characteristics of the geometry will generally have influences along the following lines: A deeper bed trough might imply a stronger ice gradient at the grounding line for similar glaciers, and thereby a stronger driving stress. In case of a MISI-type retreat, a shorter horizontal length scale, i.e., a steeper retrograde bed slope, leads to a higher flux increase for each increment of grounding-line retreat down the slope, hence amplifying the positive feedback loop between ice discharge and retreat (Schoof, 2007; Pattyn and Morlighem, 2020). A wider

confinement diminishes the buttressing strength of the system (Dupont and Alley, 2005; Gudmundsson et al., 2012; Leguy, 2015), which in turn provides less moderation to the unstable retreat.

Using the confinement geometry, the time scaling $\tau$ (Eq. 3) can be translated into a scaling of the glacier outflow $\mu$, i.e., the discharge through the confinement. For this purpose the velocity scaling $\nu = V'/V = \delta/\tau$ (based on the velocity scale $V = L/T$) is multiplied with the scaling of the confinement depth $\beta$ and width $\omega$, respectively, yielding

$$130 \quad \mu \equiv \nu \, \beta \, \omega = \frac{\delta \, \beta \, \omega}{\tau} = \alpha \, \beta^4 \, \omega^5 \, \delta^{-3} = \alpha \, \beta^4 \, \gamma^5 \, \delta^2. \tag{5}$$

This serves as a measure for the relative outflow between individual outlets, i.e., the average velocity times the vertical outflow area near the grounding line. Introducing the ratio of the depth gradients $\Gamma \equiv \beta/\delta$ this becomes

$$\mu = \alpha \, \delta \, \Gamma^4 \, \omega^5, \tag{6}$$

i.e., the outgoing volume flux scales linearly with the ice softness and the bed depth, respectively, and with the fourth power of

135 the gradient of the bed topography and with the fifth power of the confinement width. Please note that the above relations only hold under the listed assumptions, especially the assumption of geometric similarity, i.e., for glaciers that have qualitatively the same topography only in different sizes.

## 3   Testing of the scaling relation

The analytically derived time-scaling relation (Eq. 4) is tested in two very different settings. First, we analyse observational

geometric and velocity data from strongly confined outlet glaciers in Greenland. Second, we investigate the time scaling of simulated Antarctic-type MISI.

### 3.1   Observed velocity scaling of Greenland outlet glaciers

The Greenland Ice Sheet drains through many very narrow, fjord-type outlets, suggesting a particularly strong control of the bed topography on the ice-flow timescale (Morlighem et al., 2017; Catania et al., 2020). We use a dataset from Beckmann et al.

(2019) which covers twelve outlet glaciers around Greenland (Figs. 2 and S1) to compare the observed relationship between the ice-flow timescale and the confinement width of the outlets to the results predicted by theory (Eq. 3). The comparison with the theory is only valid for observed glaciers that are approximately dynamically and geometrically similar, which excludes two glaciers from the analysis (see Appendix B for details). Observations allow us to directly use the velocity scale $V = L/T$

as a time-scale measure, with velocity scaling $\nu = V'/V = \delta/\tau$. The geometry dependence of the velocity scaling according to Eq. (3) reduces to

$$\nu = \omega^4, \tag{7}$$

for $n = 3$ if differences in the width of the glaciers dominate the velocity field ($\alpha = \beta = \delta = 1$). This quartic relationship turns out to approximate the observed velocity-width scaling reasonably well (Fig. 3). Deviations from the predicted curve likely occur due to the fact that the comparison of the complex real-world glaciers breaks the underlying theoretical assumption of perfect similarity. Deviations also result from differences between the individual glaciers in their ice softness fields, in the degree of non-linearity of their ice rheologies (flow-law exponent $n$) or in the vertical and along-flow geometry of their topographic confinements, which here are all assumed to be identical.

## 3.2 Modelled MISI time scaling with idealized geometry

The scaling relation (Eq. 4) applies to the timescale of glacier *flow*, as supported by the comparison with observations of Greenland outlet glaciers of similar geometry in the previous section. This flow timescale is linked to the ice discharge across the grounding line, since a faster coastal flow, i.e., a smaller $\tau$, implies a larger grounding-line discharge $\mu$ (Eq. 5). Grounding-line retreat depends on the divergence of grounding-line discharge, which involves the flow speed at the grounding line. If we were to seek a relation for the grounding-line retreat rate, we could make the assumption that the changes in the ice-flow speed in the course of retreat are proportional the equilibrium ice-flow speed. This is a strong assumption and it might not be justified in all cases. Under this assumption the timescale of glacier *retreat* would be proportional to the timescale of glacier *flow* (see Appendix A2 for the mathematical examination). This would mean that Eq. (4) would also be applicable to the time scaling of glacier retreat.

To test this, we simulate MISI-type outlet-glacier retreat timescales under a systematic variation of each of the variables involved in Eq. (4) in an ensemble of idealized, three-dimensional numerical experiments. For this purpose we employ the Parallel Ice Sheet Model (PISM; Bueler and Brown, 2009; Winkelmann et al., 2011; Khroulev and Authors, 2020). The model solves the full SSA equations (Eq. A1) of the stress balance from which the above scaling relation is derived, thus accounting for all the horizontal compressive, tensile and shear stresses that act within the ice as well as basal shearing.

The bed topography prescribed in the simulations is obtained and modified from the MISMIP+ intercomparison project, which is the current benchmark for idealized simulations of a buttressed marine ice-sheet-shelf system (Asay-Davis et al., 2016; Cornford et al., 2020; Lhermitte et al., 2020; Zhang et al., 2020; Leguy et al., 2021; Feldmann et al., 2022). It shapes a trough with a retrograde bed section (Fig. 4), resembling the geometric characteristics of a deeply incised, MISI-prone marine Antarctic outlet such as Pine Island Glacier. Ice flow in this setup mainly takes place from the interior through the bed trough, where an ice stream feeds a bay-shaped ice shelf that calves into the ocean (Fig. S3). There is also a perpendicular flow component, i.e., from the channel's lateral ridges down into the trough. Resulting convergent flow and associated horizontal

shearing (Fig. S4) enable the emergence of buttressing. Details on the model setup and the experimental design are given in Appendix C.

In our simulations we vary separately the four relevant parameters which are found to determine the timescale of the ice dynamics according to Eq. (4). That is, the depth $D$, the width $W$ and the length $L$ of the bed trough and the ice temperature $T_{ice}$ (ice softness $A(T_{ice})$; see Eq. C1) are altered one at a time (see Table 2 for parameter ranges). The timescale $T$ is measured

as the time per km of grounding-line retreat (inverse retreat speed) down the central part of the retrograde bed section (marked grey in Fig. 5). This way we obtain four sets of simulations, i.e., for the ice-softness scaling $\alpha$, the confinement-depth scaling $\beta$, the length scaling $\delta$ and the width scaling $\omega$, each of them yielding a curve for the time scaling $\tau$ (large panels in Fig. 6). To examine the scaling behavior of the measured retreat timescales, we normalize each set of simulations with respect to the same reference softness / confinement depth / confinement length / confinement width and the associated reference timescale

(blue axes in Fig. 6) and compare the data to the theoretically expected scaling behavior according to Eq. 4 (blue curves in Fig. 6). For each scaling parameter the numerical results are in good agreement with our analytically derived time-scaling relation of Eq. (4). Linear regression of the data in double-logarithmic plots (insets in Fig. 6) yields that the scaling exponents of the geometric scales $D$, $W$ and $L$ inferred from our simulations slightly underestimate the theoretical values (deviation by $\approx 13\%$). Note that the theoretical scaling relation assumes negligible basal stresses, while in our simulations the basal stresses

are not negligible (due to the prescribed non-negligible value of the basal friction coefficient $C$ in the sliding law), which is likely one reason for the deviations between simulations and theory. Additional simulations that vary the friction coefficient over one order of magnitude show that the retreat speed increases with decreasing friction with the time scaling following Eq. (4) to a good approximation (Fig. S5). Deviations from theory increase with decreasing friction since reducing $C$ flattens the originally parabolic glacier profile from the reference simulation, leading to a violation of the similarity principle.

An additional set of experiments of altered accumulation rate $a$ is conducted to analyze how a scaling of the surface mass balance affects the time scaling of grounding line movement. The simulations yield a slightly larger or smaller ice sheet depending on whether the accumulation rate is increased or decreased. This in turn slightly changes the depth, length and width measurements of our analysis and thereby the timescale. Figure S5 shows that the changes induced by the relative accumulation changes have a relatively weak influence on the timescale.

**4 Application of scaling relation to Antarctic outlet glaciers**

Our analysis of observed Greenland outlet-glacier velocities revealed that their scaling follows the theoretical prediction from the scaling relation (Sect. 3.1; Fig. 3). The validity of the scaling relation is further supported by the conducted ensemble of idealized, numerical simulations of Antarctic-type outlet-glacier retreat (Sect. 3.2), as simulated MISI retreat timescales are in good agreement with the ones predicted from theory (Fig. 6). This leads us to apply the theory to observations of Antarctica's

MISI-prone outlet glaciers to compare their retreat timescales after a potential destabilization. We base our approach on the fact that the currently observed geometry and physical conditions of the considered outlet glaciers represent the balance of the

relevant forces that act on the ice. This static situation carries the information for their initial retreat timescale after a potential destabilization.

Analyzing the major outlets around Antarctica, we find 13 glaciers that show a clear retrograde bed slope upstream of their grounding line and have a generally similar bed profile. Please refer to the Supplementary Figures S9-S23 for maps of the topographic features of each glacier and Appendix D1 for a definition of the applied similarity criteria. These glaciers are situated in the Amundsen Sea Sector, East Antarctica and the Filchner-Ronne region (Fig. 7). The three geometric scales required by Eq. (4), i.e., the depth, width and length scales are obtained from the recent BedMachine Antarctica compilation (Morlighem et al., 2020). The ice softness (Fig. S9), entering the scaling relation as the fourth parameter, is obtained from a present-day simulation of the Antarctic Ice Sheet, which was conducted in the course of the recent ISMIP 6 Antarctica intercomparison project (Seroussi et al., 2020). Pine Island Glacier (PIG) is selected as the scaling reference for our analysis as one of the most studied glaciers in Antarctica. The scales are extracted along the streamlines of each outlet (Figs. S10-S24, panels A,C and D), based on the MEaSUREs Antarctic velocity map from Mouginot et al. (2019). From this we obtain probability distributions of the respective scaling ratios (Figs. S10-S24, panel E), representing the spatial variability of the extracted data. The distributions of the retreat-time scaling (Fig. 8) are calculated according to Eq. (4) using a Monte Carlo method. The medians of these distributions, $<\tau>$, represent the best estimate of our calculations. Details on our method and the underlying datasets are given in Appendix D.

The obtained timescales range over three orders of magnitude (Fig. 7 and Table 3), mainly due to differences in the bed-slope magnitude (length scale), the width of the confinement and its depth, as these scales enter Eq. (4) by the power of 3 and 4, respectively. Below we will refer to the inverse values of the time-scaling estimates, i.e., $<\tau>^{-1}$, providing a measure of the glaciers' potential retreat speed relative to the reference PIG.

By far the fastest retreat speed after a potential destabilization is detected for Mellor Glacier (MG; $<\tau_{MG}>^{-1} = 67$). Here the slowing influence of its relatively narrow confinement is strongly overcompensated by the combination of a deep trough (about 2000 m; together with Foundation Ice Stream (FIS) the deepest of the ensemble, see Table 3 and Fig. S8) and a steep retrograde bed slope (with about -55 m/km the steepest of the entire ensemble). The fast retreat speed of Ninnis Glacier (NG; $<\tau_{NG}>^{-1} = 20$) is primarily caused by its relatively steep retrograde slope. Similar retreat rates are obtained for the outlets of Thwaites Glacier (TG; $<\tau_{TG}>^{-1} = 19$) and Cook Glacier (CG; $<\tau_{CG}>^{-1} = 16$), which is for different reasons: While CG has the widest confinement among all investigated outlets (100 km), being almost twice as wide as TG's confinement, its retrograde slope is only half as steep compared to the slope of TG. Both effects roughly balance out and since the scaling ratios for the ice softness and the trough depth are very similar, respectively, this leads to the similar timescale. The reference glacier of our analysis, PIG, is found to have a much slower retreat speed. The effect of having the second softest ice of all examined outlets is outdone by the glacier's comparatively gentle slope and narrow trough, respectively. Slower retreat rates are only found for Recovery Glacier (RG; $<\tau_{RG}>^{-1} = 0.8$; slightly narrower, flatter bed trough) and FIS ($<\tau_{FIS}>^{-1} = 0.1$; much flatter slope). Smith Glacier and Denman Glacier show a pronounced bimodal time-scale distribution (highlighted in

Fig. 8) which is an imprint of their bimodal distributions: for both glaciers the inferred depth scales $D$ span a relatively large range and cluster around the two endings of it, revealing two distinct depth regimes at the grounding line (Figs. S11 and S14). For some glaciers the likely ranges of $\tau$ are very large, which is a combined effect of 1) our method of extracting the characteristic scales, capturing the full spatial variability inside the confinement and 2) the high degree of non-linearity of the scaling relation, amplifying the uncertainty in the obtained scaling ratios. Note that we do not account for the uncertainty of the

underlying topographic data itself. As the density of measurements is relatively high over many of the Antarctic outlet glaciers, the vertical accuracy over the regions considered in this study is typically on the order of $\sim 10$ m, lying at the very low end of the overall Antarctic uncertainty range (Morlighem et al., 2020).

Despite having the fastest timescale, the sea-level contribution of a potential retreat of MG could be comparatively small (Fig. 7). For instance, the volume of marine ice in the drainage basin of Slessor Glacier (SLG) / RG and CG / NG is more than

255 2.5 times as large compared to the basin of MG. The consequences of a destabilization could thus be more serious for CG, NG and Support Force Glacier (SFG) as these tributaries show both a high retreat speed and a large sea-level potential.

## 5    Discussion and conclusions

Exploiting the similitude principle, we derived a time-scaling relation for the case of fast, shallow, and laterally confined outlet-glacier flow with negligible basal friction (Eq. 4). According to this relation, the timescale of an outlet is proportional to the

260 inverse of its ice softness and highly non-linearly related to the width, length and depth scales of its confinement. That is, deeper and wider outlets with a steeper retrograde slope have a faster timescale than shallower, more confined outlets with a flatter slope (Fig. 1). We showed that the derived scaling relation is able to predict the scaling of 1) observed flow velocities of Greenland outlet glaciers (Sect. 3.1; Fig. 2) and 2) Antarctic-type MISI retreat speed as simulated in idealized experiments (Sect. 3.2; Fig. 6). Motivated by these simulations, we applied the scaling relation to the observed topography of 13 MISI-prone

Antarctic outlets to compare their time scaling of potential unstable retreat (Sect. 4; Fig. 7). Within this set of outlets, Mellor, Ninnis, Thwaites, Cook Glaciers have the fastest timescales - about 16 to 67 times faster than the reference Pine Island Glacier (Fig. 8; Table 3) - due to their particularly wide and/or deep confinements and/or steep slopes (Figs. S8 and S10-S24). The underlying approach and its application have several limitations which restrict the results with respect to:

1. **Similitude requirement:** The application of Eq. (4) to observed glaciers is limited by the requirement of geometric
similarity. That is to say that the approach can only be used to compare existing outlet glaciers if they can be scaled onto each other as shown in the Supplementary figures (Figs. S10-S24, panel A). Effects of differences in ice-shelf buttressing or smaller topographic differences such as ice rises and details in the bed topography can alter the timescale of different glaciers in a way that cannot be captured by the approach because it weakens the similitude assumption. Perfect geometric similarity between two systems can only be achieved in a synthetic environment, i.e., in idealized
experiments such as the ones carried out in this study to test the scaling relation (Sect. 3.2). The results from our

numerical simulations suggest that the scaling relation is valid for strictly similar geometries (Fig. 6). The real-world glaciers considered here show geometric similarity in a broader sense.

2. **Region of applicability:** Our analysis is restricted to regions in which basal shear stresses are small compared to the ice-internal stresses (including the buttressing-inducing lateral shear stresses). This assumption generally holds for coastal regions of most Antarctic outlet glaciers where basal shear stresses are found to be close to zero (Joughin et al., 2004; Morlighem et al., 2013; Sergienko and Hindmarsh, 2013; Sergienko et al., 2014) - with the possibility of the sporadic occurrence of confined and comparatively small rib-like patterns of high basal resistance in some areas (Sergienko and Hindmarsh, 2013; Sergienko et al., 2014). The region of low basal friction is likely to follow the grounding line in case of retreat as the lubrication of the bed is largely caused by the abundance of liquid water at the base of the ice sheet (Schoof, 2005; Martin et al., 2011; Leguy et al., 2014).

3. **Time period of applicability:** In principle the applicability of our analysis is also limited by possible changes in surface mass balance. These are, however, likely slower than the ice-dynamic discharge. Estimates of the past decades show a background snowfall in coastal regions of at most $1$ m/yr (Wessem et al., 2014; Agosta et al., 2019). Changes in this quantity can be estimated by a $5 - 7$ % increase for every degree of warming in the atmosphere (Frieler et al., 2015) and are thus on the order of $0.05 - 0.07$ m/yr/K. Differences in the surface mass balance between individual outlets are modeled to be on the order of $0.1$ m/yr (Wessem et al., 2014; Agosta et al., 2019). In contrast, the highest observed dynamic thinning rates of the destabilized outlets in West Antarctica's Amundsen Sea Sector typically reach up to $5 - 10$ m/yr (Pritchard et al., 2009; Konrad et al., 2018; Shepherd et al., 2019), making changes in the surface mass balance a second-order effect with respect to the retreat timescale. Observed average grounding-line retreat rates of these outlets reach up to $1$ km/yr (Rignot et al., 2014; Konrad et al., 2018) from which we estimate that our results may be valid for at least two decades after a potential destabilization. After that the analysis might have to be updated with the altered topography. A comparison of our numerical scaling results (insets of Fig. 6) to an additional set of simulations of altered surface accumulation yields that a scaling of the surface mass balance has a relatively weak influence on the time scaling of grounding-line retreat (Fig. S6).

4. **Relation between timescales of ice flow and retreat:** The theoretical scaling relation is derived for the time scaling of the ice flow, e.g., the scaling of the flow speed of similar outlet glacier as examined in Sect. 3.1. We use the same equation to compare the time scaling of grounding line retreat, which requires that the timescales of ice flow and retreat, $T_{\mathrm{flow}}$ and $T_{\mathrm{retr}}$, are proportional to each other. This proportionality is given under the assumption that changes in the ice-flow speed due to retreat are proportional to the ice-flow speed in equilibrium (as shown in Appendix A2). Although confirmed in the idealized simulations (Sect. 3.2), this assumption generally puts further constraints on the applicability of Eq. (4) to the potential retreat timescales of real-world glaciers.

Within these very strong constraints the scaling relation can capture dependencies of highly complex stress- and flow fields. It resolves the buttressing effect of the grounded portion of an outlet explicitly, by extracting its dependence on the geometry of

the outlets' grounded lateral confinement. In our analysis we infer the associated characteristic scales upstream of the grounding
line focusing on the retrograde bed section as the crucial region that determines the timescale of a potential instability and thus
we do not take into account the ice-shelf geometry. In our simulations the ice shelves and the upstream confinement share the
same width scale (Figs. 4 and S3). The outlets feeding the large Filchner-Ronne Ice Shelf deviate from this assumption. They
might respond on a slower timescale than inferred here due to the stabilizing influence of the large ice rises and other smaller
pinning points in the central region of the ice shelf and the convergent flow of at least ten ice streams that enter the ice shelf,
effects that cannot be accounted for by our approach.

For the majority of outlets analyzed here, a comparison of the bed topography and the ice-flow pattern suggests a strong
control of the bed topography on the width of the outlet (e.g., Pine Island Glacier, Denman Glacier, Mellor Glacier or Slessor
Glacier; Figs. S10, S14, S16, S17; visualized schematically in Fig. 1). That is, the lateral margins of these glaciers are clearly
defined by the lateral ridges of the underlying bed trough. However, a few glaciers (e.g., Thwaites Glacier and Ninnis Glacier;
Figs. S10 and S13) reveal a much weaker topographic control. We base the delineation of the outlets' lateral margins on the
$e$-folding scale of their velocity field. While other methods of deducing the glacier-width scale may lead to different results in
cases of weak topographic influence, our approach provides a well-defined width measure that can be applied consistently to
all outlets, independent of their individual degree of topographic control.

Outlets discarded in this study include Pope Glacier and Kohler Glacier (eastern and western branch). Though both troughs of
Kohler Glacier have a retrograde bed section upstream of the grounding line, their strongly undulating bed topography further
upstream (Figs. S22 and S23) deviates substantially from the rest of the ensemble. Pope Glacier (Fig. S24) is excluded from our
analysis due to its very short retrograde slope section (3 km). The main outlets feeding Ross Ice Shelf, e.g., Bindschadler Ice
Stream and MacAyeal Ice Stream, and also Totten Glacier in East Antarctica, are found to have a generally, i.e. flux-weighted,
flat or prograde bed topography upstream of the grounding line.

The timescales obtained here differ from the ones estimated in a previous study in which we also applied the similitude
principle to Antarctic outlets (Levermann and Feldmann, 2019). This is due to the fact that the two studies make different
assumptions in the dimensional analysis of the SSA leading to different scaling relations which yield different results. For
instance, the one-dimensional analysis in Levermann and Feldmann (2019) does not account for the across-flow direction ($y$
direction here) and thus neglects the effect of horizontal shearing. In the present study horizontal shear stresses turn out to
represent the most important stresses compared to all other ice-internal stresses, originating from the assumption of a generally
small width-to-length ratio of Antarctica's outlet glaciers. In fact, the timescale of TG estimated here is about ten times faster
than in the previous study, since the present study takes into account the much larger width of TG (and thus the smaller
buttressing effect of its grounded portion) compared to the reference PIG. In contrast, in the present study basal stresses are
neglected, whereas in Levermann and Feldmann (2019) there is a separate scaling law that involves basal friction parameter.
Furthermore, the present study uses a more sophisticated method of extracting the outlets' characteristic scales, i.e., they are
extracted for each glacier along a set of streamlines spanning its confinement compared to a single linear central profile used

in Levermann and Feldmann (2019). The method of extraction also has an influence on which glaciers are labeled as MISI prone, such that the ensembles of outlets for which we obtain time scaling values presented here and in our previous work are not the same. Such qualitative differences between the two studies naturally lead to different timescale estimates which should
be interpreted in the light of the underlying assumptions.

The scaling approach presented here provides a useful tool that in particular allows to compare (MISI) timescales between similar glaciers within the limits of the underlying assumptions and constraints - without the need of computationally expensive numerical simulations. Further studies are required in particular to explore the role of ice-shelf buttressing on the time scaling, allowing for a more comprehensive scaling analysis.

*Code and data availability.* Antarctic bed topography and ice thickness are taken from the Bedmachine Antarctica compilation (MEaSUREs BedMachine Antarctica, Version 1) and can be downloaded from https://nsidc.org/data/NSIDC-0756/versions/1. Antarctic surface velocities (MEaSUREs Phase-Based Antarctica Ice Velocity Map, Version 1) are available at https://nsidc.org/data/NSIDC-0754/versions/1. The simulation data generated and analyzed in this study are available from https://doi.org/10.5281/zenodo.7395830 (Feldmann, 2022). PISM is freely available as open-source code from https://github.com/pism/pism. The code version used in this study can be obtained from
https://doi.org/10.5281/zenodo.7395020 (Feldmann and PISM authors, 2022).

### Appendix A: Similitude analysis of shallow ice dynamics

#### A1 Timescales of ice flow

We apply the concept of similitude (Buckingham, 1914; Rayleigh, 1915; Szücs, 1980) to the dynamics of ice sheets based on the shallow-shelf approximation (SSA; Morland, 1987; MacAyeal, 1989) of the Stokes stress balance (e.g., Le Meur et al.,
2004). Neglecting the terms of vertical shearing in the stress balance and accounting for the small thickness-to-length ratio of ice sheets, the SSA represents the relevant dynamics of floating ice shelves and grounded ice streams, characterized by fast, plug-like flow. In particular, we assume isothermal ice, a condition that has been used to analyze ice-sheet dynamics in a number of previous studies (e.g., Dupont and Alley, 2005; Goldberg et al., 2009; Gudmundsson et al., 2012; Pattyn et al., 2013; Feldmann and Levermann, 2016; Seroussi and Morlighem, 2018; Cornford et al., 2020).

For an ice stream with horizontal velocity components $v_x$ and $v_y$, ice thickness $H$, ice surface elevation $h$, and spatially uniform (isothermal) ice softness $A$, the SSA has the following two horizontal components,

$$
\begin{aligned}
A^{-1/n}\left(\frac{\partial}{\partial x}\left[H\dot{\epsilon}_e^{1/n-1}\left(2\frac{\partial v_x}{\partial x}+\frac{\partial v_y}{\partial y}\right)\right]+\frac{1}{2}\frac{\partial}{\partial y}\left[H\dot{\epsilon}_e^{1/n-1}\left(\frac{\partial v_x}{\partial y}+\frac{\partial v_y}{\partial x}\right)\right]\right)+\tau_{b,x}=\rho g H\frac{\partial h}{\partial x}, \\
A^{-1/n}\left(\frac{\partial}{\partial y}\left[H\dot{\epsilon}_e^{1/n-1}\left(2\frac{\partial v_y}{\partial y}+\frac{\partial v_x}{\partial x}\right)\right]+\frac{1}{2}\frac{\partial}{\partial x}\left[H\dot{\epsilon}_e^{1/n-1}\left(\frac{\partial v_y}{\partial x}+\frac{\partial v_x}{\partial y}\right)\right]\right)+\tau_{b,y}=\rho g H\frac{\partial h}{\partial y},
\end{aligned}
\tag{A1}
$$

where $\dot{\epsilon}_e$ is the effective strain rate, $\tau_b$ is the basal shear stress, $\rho$ is the ice density, $g$ the gravitational acceleration and $n$ denotes Glen's flow-law exponent (Glen, 1955). The SSA implies that the horizontal velocities do not vary in vertical ($z$) direction.

Eqs. (A1) describe the balance between 1) ice-internal stresses due to shearing, extension and compression (in large brackets), 2) the basal shear stress $\boldsymbol{\tau}_b = (\tau_{b,x}, \tau_{b,y})$ between ice and bed and 3) the driving stress due to the gradient in the ice-surface elevation. Here we consider the case in which the driving stress is mainly balanced by the ice-internal stresses and we will neglect the basal shear stress term in the following.

The effective strain rate $\dot{\epsilon}_e$ in Eqs. (A1) can be written as

$$\dot{\epsilon}_e = \left[ \left( \frac{\partial v_x}{\partial x} \right)^2 + \left( \frac{\partial v_y}{\partial y} \right)^2 + \frac{\partial v_x}{\partial x} \frac{\partial v_y}{\partial y} + \frac{1}{4} \left( \frac{\partial v_x}{\partial y} + \frac{\partial v_y}{\partial x} \right)^2 \right]^{1/2}. \tag{A2}$$

Introducing the geometric scales $S_x$ and $S_y$ as the two characteristic horizontal dimensions of the system and a timescale $T$, we bring the components of the effective strain rate (Eq. A2) into dimensionless form (Buckingham, 1914) via $x^* = \frac{x}{S_x}$, $y^* = \frac{y}{S_y}$, $v_x^* = \frac{v_x T}{S_x}$ and $v_y^* = \frac{v_y T}{S_y}$, which yields

$$\dot{\epsilon}_e = T^{-1} \left[ \left( \frac{\partial v_x^*}{\partial x^*} \right)^2 + \left( \frac{\partial v_y^*}{\partial y^*} \right)^2 + \frac{\partial v_x^*}{\partial x^*} \frac{\partial v_y^*}{\partial y^*} + \frac{1}{2} \frac{\partial v_x^*}{\partial y^*} \frac{\partial v_y^*}{\partial x^*} + \frac{1}{4R^2} \left( \frac{\partial v_x^*}{\partial y^*} \right)^2 + \frac{R^2}{4} \left( \frac{\partial v_y^*}{\partial x^*} \right)^2 \right]^{1/2}, \tag{A3}$$

with

$$R = \frac{S_y}{S_x}. \tag{A4}$$

Defining the dimensionless variables $H^* = \frac{H}{D}$ and $h^* = \frac{h}{D}$ with the vertical geometric scale $D$ we now non-dimensionalize the $x$-component of the SSA (Eqs. A1), writing

$$\frac{K}{T} \frac{\partial}{\partial x^*} \left[ 2H^* \dot{\epsilon}_e^{1/n-1} \frac{\partial v_x^*}{\partial x^*} + H^* \dot{\epsilon}_e^{1/n-1} \frac{\partial v_y^*}{\partial y^*} \right]$$
$$+ \frac{K}{2T} \frac{\partial}{\partial y^*} \left[ H^* \dot{\epsilon}_e^{1/n-1} \frac{1}{R^2} \frac{\partial v_x^*}{\partial y^*} + H^* \dot{\epsilon}_e^{1/n-1} \frac{\partial v_y^*}{\partial x^*} \right] = H^* \frac{\partial h^*}{\partial x^*}, \tag{A5}$$

with

$$K = \frac{1}{\rho g A^{1/n} D}. \tag{A6}$$

We specify that the main flow direction, i.e., along the topographic confinement, is in $x$ direction and that the transversal, across-trough, flow component is in $y$ direction. In this case, the two geometric horizontal scales can be associated with a length scale $L = S_x$ and a width scale $W = S_y$ of the confinement. Observations indicate that many of the outlet glaciers of

390 Greenland and Antarctica stream through bed confinements with a characteristic length that is several times (in places one order of magnitude) larger than the confinement width. Consequently, we assume $L \gg W$ and thus $R \ll 1$.

Accounting for the leading order term in the effective strain rate (Eq. A3), which is on the order of $\mathcal{O}(1/R)$ (using Landau notation, e.g., Hardy and Wright, 1979), the effective strain rate can be approximated by

$$\dot{\epsilon}_e \approx \frac{1}{2RT} \frac{\partial v_x^*}{\partial y^*}. \tag{A7}$$

Analogous, in the SSA (Eq. A5) only the $1/R^2$ term remains on the left-hand-side of the equation. Using Eq. (A7), we can eventually express the approximated SSA as

$$\lambda \frac{\partial}{\partial y^*} \left[ H^* \left( \frac{\partial v_x^*}{\partial y^*} \right)^{1/n} \right] = H^* \frac{\partial h^*}{\partial x^*}, \tag{A8}$$

with

$$\lambda = \frac{K}{2^{1/n} T^{1/n} R^{1/n+1}}. \tag{A9}$$

From Eq. (A8), we see that the stress balance remains exactly the same as long as the dimensionless coefficient $\lambda$ remains the same. In other words, the first-order ice-sheet dynamics of our problem are expected to be similar under a transformation that leaves this coefficient unchanged. Applying a scaling to the system (denoted by a dash) and requiring similarity ($\lambda' = \lambda$) yields

$$\frac{1}{A' D'^n T'} \left( \frac{L'}{W'} \right)^{n+1} = \frac{1}{A D^n T} \left( \frac{L}{W} \right)^{n+1}. \tag{A10}$$

Introducing the scaling ratios $\alpha = \frac{A'}{A}$, $\beta = \frac{D'}{D}$, $\omega = \frac{W'}{W}$, $\delta = \frac{L'}{L}$ and $\tau = \frac{T'}{T}$ for the geometric dimensions, the ice softness and the timescale of the system, respectively, we arrive at

$$\tau = \alpha^{-1} \beta^{-n} \left( \frac{\omega}{\delta} \right)^{-(n+1)}. \tag{A11}$$

Since $n$ is always positive, the timescale of the system will increase ($\tau > 1$) for a reduction of the ice-softness scale ($\alpha < 1$), a shortening of the vertical geometry ($\beta < 1$), an elongation of the geometry in main-flow direction ($\delta > 1$), or a narrowing
of the across-main-flow geometry ($\omega < 1$). Defining $\gamma = R'/R = \omega/\delta$ and making the common choice of $n = 3$ (Glen, 1955; Duval, 2013) we eventually obtain

$$\tau = \alpha^{-1} \beta^{-3} \gamma^{-4}. \tag{A12}$$

## A2 Relation between timescales of ice flow and grounding-line retreat

Note that Eq. (A12) refers to the timescale of glacier flow, $T_{\text{flow}}$, which is associated with the velocity of ice flow in equilibrium
(see Appendix B). In order to find a relation of this timescale to the timescale of grounding-line retreat, $T_{\text{retr}}$, in the following we analyze the time evolution of a change in the ice thickness close to the grounding line of a perturbed outlet. We define this change in ice thickness as

$$\delta H = H - H_{\text{eq}}, \tag{A13}$$

where $H_{\text{eq}}$ is the ice thickness in equilibrium and $H$ is the ice thickness in the perturbed state. The time evolution of $\delta H$ then reads

$$\frac{\partial(\delta H)}{\partial t} = \frac{\partial H}{\partial t}, \tag{A14}$$

since $\frac{\partial H_{\text{eq}}}{\partial t} = 0$ per definition. We now introduce the ice thickness equation (e.g., **?**) which relates changes in the ice thickness to the divergence of the ice flux $Q$ and the surface accumulation $a$

$$\frac{\partial H}{\partial t} = -\nabla \cdot Q + a. \tag{A15}$$

Accounting for the fact that the fast retreat of Antarctic outlets such Pine Island Glacier and Thwaites Glacier is due to their ice-dynamic imbalance (Pritchard et al., 2009; Konrad et al., 2018; Shepherd et al., 2019), i.e., due to changes in the ice flux, we can neglect changes in $a$, and write

$$\frac{\partial H}{\partial t} = \frac{\partial(\delta H)}{\partial t} = -\nabla \cdot (\delta(Hu)), \tag{A16}$$

where we used $Q = Hu$ with the ice-flow velocity $u$ in equilibrium. Note that we used $\delta \nabla \cdot (Hu) = \nabla \cdot \delta(Hu)$ due to the linearity of the nabla operator $\nabla$. We now write Eq. (A16) in full, which yields

$$\frac{\partial \delta H}{\partial t} = -(\nabla \cdot u)\delta H - (\nabla \cdot \delta H)u - (\nabla \cdot \delta u)H - (\nabla \cdot H)\delta u. \tag{A17}$$

If we make the plausible assumption that the equilibrium velocity profile increases in downstream direction, i.e., $\nabla \cdot u > 0$, then the first term on the right-hand-side of Eq. (A17) has a stabilizing character. That is, the change $\delta H$ and its temporal evolution, $\frac{\partial \delta H}{\partial t}$, have opposite signs such that the change is dampened over time. Although this is an important term in the general case, we neglect it here, since it cannot be the decisive term in the case of self-sustained grounding-line retreat, which we analyze here. If we consider the spatial derivatives of $\delta H$ and $\delta u$ to be of second order, then Eq. (A17), in combination with Eq. (A14), reduces to

$$\frac{\partial H}{\partial t} = -(\nabla \cdot H)\delta u. \tag{A18}$$

Writing down the dimensions of Eq. (A18), its left-hand-side reads

$$\left[\frac{\partial H}{\partial t}\right] = \frac{D}{T_{\text{retr}}}, \tag{A19}$$

with ice-thickness scale $D$ and timescale of retreat $T_{\text{retr}}$, which is associated with the thinning rate. The right-hand-side of Eq. (A18) yields

$$[(\nabla \cdot H)\delta u] = \frac{D}{L}[\delta u], \tag{A20}$$

with length scale $L$. To find an expression for $[\delta u]$ we make the assumption that the characteristic scale of a change in the ice-flow velocity is proportional to the ice-flow velocity itself, i.e.,

$$[\delta u] \sim [u] = \frac{L}{T_{\text{flow}}} \tag{A21}$$

where $T_{\text{flow}}$ is the timescale of the equilibrium ice-flow velocity. The dimensional analysis of Eq. (A18) thus yields

$$\frac{D}{T_{\text{retr}}} \sim \frac{D}{T_{\text{flow}}}. \tag{A22}$$

Considering a scaled (dashed) and an unscaled system with $\tau_{\text{retr}} = \frac{T'_{\text{retr}}}{T_{\text{retr}}}$ and $\tau_{\text{flow}} = \frac{T'_{\text{flow}}}{T_{\text{flow}}}$ it follows

$$\tau_{\text{retr}} = \tau_{\text{flow}}, \tag{A23}$$

i.e., the time scaling of grounding-line retreat is equal to the time scaling of ice flow.

## Appendix B: Analysis of Greenland outlet glaciers

The observational data used in our analysis is obtained from Beckmann et al. (2019) who inferred along-flow profiles of the bed elevation, ice thickness, ice velocity and glacier width for a selection of Greenland outlet glaciers based on flux-weighted
averages using geometric data from Morlighem et al. (2014) and velocity data from Rignot and Mouginot (2012). The twelve glaciers cover a variety of sizes, locations, discharge rates and climatic conditions across the Greenland Ice Sheet (Beckmann et al., 2019). At the same time most of these outlets are qualitatively similar to each other as their flow is strongly confined by a narrow lateral bed geometry and they are grounded on bedrock that is generally sloping up in landward direction (Figs. 2 and S1). Two glaciers deviating from these conditions, i.e., Daugaard-Jensen Glacier and Rink Glacier, are excluded from our
analysis. Daugaard-Jensen Glacier is the only outlet in the ensemble that is grounded on a landward down-sloping (instead of up-sloping) bed topography. It has been shown that these two types of bed slope can have fundamentally different effects on the ice-flow dynamics (Schoof, 2007). Rink Glacier exhibits a pronounced inversion of the ice-flow acceleration near the terminus, which might be due to a local specific topographic feature, making the outlet's ice-speed profile qualitatively very different compared to the rest of the outlets (except for Upernavik Isstrom N). Since the inversion for the case of Upernavik
Isstrom N is much less pronounced, shorter in extent and closer to the calving front we decided to include this glacier in our analysis. In fact, these choices involve some degree of subjectivity and a different approach of ensuring qualitative similarity between the glaciers might lead to a different set of valid glaciers to be analyzed.

To extract a characteristic scale for the confinement width $W$ of each of the individual glaciers we average the width profile over the first $60\,\text{km}$ upstream of the glacier terminus. The same is done for the velocity scale $V$, though here leaving out the
470 first $20\,\text{km}$ to exclude the possible influence of ice-ocean interactions, i.e., melting and calving events, and their effects near the glacier terminus. The analysis of Gade Glacier, with its relatively short catchment, is limited to the first $40\,\text{km}$ upstream of its terminus.

With the flow-speed scale $V = L/T$, the speed scaling ratio $\nu = V'/V$ can be written as

$$\nu = \frac{L'\,T}{L\,T'} = \delta\,\tau^{-1}. \tag{B1}$$

Assuming that both the time scaling and the velocity scaling of the considered glaciers are dominated by the width of their narrow topographic confinements ($\alpha = \beta = \delta = 1$) the analytically derived scaling relation (Eq. 4) reduces to

$$\tau = \omega^{-4}, \tag{B2}$$

and consequently

$$\nu = \omega^4, \tag{B3}$$

which provides a good approximation to the scaling behavior deduced from the observational data (Fig. 2). Note that the results are affected by the concrete choice of averaging ranges along the glaciers' velocity and width profiles from which the scales of $V$ and $W$ are estimated (Fig. S2). For instance, taking the full range of the velocity profile to calculate the velocity average instead of leaving out the ocean-ward third of the profile leads to larger deviations from the theoretically predicted scaling curve (Fig. S2, compare panels A/B to C/D).

**Appendix C: Idealized simulations of MISI-type retreat**

We apply PISM to a modified version of the MISMIP+ setup to simulate the unstable retreat of an inherently buttressed Antarctic-type outlet glacier (Figs. 4 and S3). In the default MISMIP+ setting the model spinup yields a steady-state outlet glacier with a stable grounding-line position on the retrograde slope section of the bed (Asay-Davis et al., 2016; Cornford et al., 2020). A perturbation of this equilibrium leads to a reversible retreat of the outlet. That is, the triggered transient retreat stops and the grounding line migrates back toward its original equilibrium location once the perturbation ceases, even for strong perturbation magnitudes of basal ice-shelf melting or a reduction of the basal resistance (Gudmundsson et al., 2012; Cornford et al., 2020). In order to be able to extract timescales of irreversible, i.e., MISI-type retreat, we widened and deepened the bed trough of the MISMIP+ setup. For this purpose we doubled the extent of the computational domain in $y$ direction (320 km instead of 160 km) and also extended it in $x$ direction (800 km instead of 700 km). In MISMIP+ the bed topography results from the superposition of the two components $B_x(x)$ and $B_y(y)$, representing the two horizontal dimensions. In our experiments the $y$ component, $B_y(y)$, is the same as in the MISMIP+ setup (Cornford et al., 2020, Eq. 1) but we generally apply larger values for the channel width $w_c$ troughout our experiments (see Table 2 for the range of used values). The $x$ component, $B_x(x)$, is qualitatively very similar to the one used in MISMIP+ but it is generated using a piecewise cubic spline interpolation instead of prescribing a polynomial. This allows for a convenient prescription of the position and elevation of the endpoints of the retrograde bed section: at the nodes $x_0 = 0$ (location of the inland summit), $x_1 = 200$ km and $x_2 = 500$ km (landward and oceanward ends of the retrograde bed section), and $x_3 = 700$ km we prescribe the bed elevation ($B_x(x_0) = -400$ m, $B_x(x_1) = -750$ m, $B_x(x_2) = -600$ m and $B_x(x_3) = -800$ m) and set its first derivative to zero (since the nodes are locations of local extrema). Beyond $x_3$ the bed elevation is set to $B_x(x > x_3) = -800$ m. The shape of $B_x(x)$ resulting from the interpolation is shown in Fig. 4B.

For a sufficiently wide trough, the outlet glacier that evolves after model initialization retreats down the entire retrograde slope, finding a stable equilibrium only on the landward up-sloping bed section close to the ice divide (reddish lines in Fig. 5).

Using this experimental design, all of our conducted simulations share the same initial conditions (except for the varied parameter) and thus have a high degree of similarity, which facilitates the application of the scaling to our numerical results.

The experiments are initiated from a block of ice of $2,000$ m thickness and are run into equilibrium for several $10,000$ model years. Ice surface accumulation due to snowfall takes place at a spatially uniform rate ($a = 0.15$ m/yr) and is constant in time. Basal and surface melting of the ice are neglected. The temperature within the ice body, $T_{\mathrm{ice}}$, is chosen to be uniform (isothermal) and is translated into an ice-softness value $A(T_{\mathrm{ice}})$ by an Arrhenius law (Glen, 1955) of the form

$$A(T_{\mathrm{ice}}) = A_0 e^{-Q/R_g T_{\mathrm{ice}}}, \tag{C1}$$

with constants $A_0$, $Q$ and $R_r$ (see Table 2).

For the grounded portion of the ice sheet the basal shear stress in Eqs. (A1), $\boldsymbol{\tau}_b = (\tau_{b,x}, \tau_{b,y})$, is given by a Weertman-type sliding law (e.g., Pattyn et al., 2013),

$$\boldsymbol{\tau}_b = -C|\boldsymbol{v}|^{m-1}\boldsymbol{v}, \tag{C2}$$

with constant friction coefficient $C$ and basal sliding exponent $m$. The parameter values used in the simulations are given in Table 2. Like in several earlier studies, which involved simulations of ice-sheet dynamics in idealized settings (e.g., Feldmann and Levermann, 2015, 2016; Feldmann et al., 2022), we interpolate the basal friction at the gronding line according to a sub-grid interpolation of the grounding line position (Gladstone et al., 2010; Feldmann et al., 2014). This approach attenuates the discontinuity in the basal stresses when going from the last grounded (non-zero basal friction) into the first floating cell (zero basal friction). Note that more complex sliding laws, such as a Coulomb-limited law, intrinsically ensure that the basal stresses approach zero within a transition zone at the grounding line. The application of the Coulomb-limited law from Tsai et al. (2015) in PISM in the course of the MISMIP+ intercomparison exercise (Cornford et al., 2020) revealed a narrow transition zone of $2$ km (equal to the horizontal resolution used here). Thus, in both model realizations (Coulomb-limited law vs. Weertman law with friction interpolation), the drop in basal stress takes place over the same short distance of $2$ km upstream of the grounding line. Both model realizations lead to qualitatively the same response to the applied perturbations in the MISMIP+ experiments, with moderate deviations in the initial grounding-line position and the response magnitude (Cornford et al., 2020).

The simulations are carried out on a regular horizontal grid of $2$ km resolution, which assures adequate accuracy in modeling the rapid ice dynamics, as demonstrated by a convergence study (Fig. S7). In PISM, the grounding lines are diagnosed via the flotation criterion and thus evolve freely. Grounding line movement has been evaluated in the model intercomparison projects MISMIP3d (Pattyn et al., 2013; Feldmann et al., 2014) and MISMIP+ (Cornford et al., 2020).

## Appendix D: Measuring the Antarctic outlets

### D1 Method

To apply the above derived theory to Antarctica's outlet glaciers, their characteristic scales involved in Eq. (4) have to be extracted from observations. To this end, for each outlet first we define a grounding-line section that spans the glacier's main

trunk, i.e., the part of the grounding line across which the vast majority of the glacier's total ice discharge takes place. Then, based on the velocity field of the ice, we infer the glacier's streamlines that meet this grounding-line section. The sampling resolution between and along the streamlines is 1 km. The flux-weighted average of the outlet's bed profile over all streamlines provides a general start point ($p_1$) and end point ($p_2$) of its retrograde bed section directly upstream of the grounding line (exemplary indicated in Fig. S10). For each streamline the confinement depth $D$ is then measured as the depth of the bed below zero at location $p_1$. The length scale $L$ of the retrograde bed section is obtained from the bed-slope magnitude $S$ via $L = 1/S$ for each of the streamlines that have a negative mean slope between $p_1$ and $p_2$. The confinement width $W$ is measured in perpendicular direction to a central streamline of the outlet (indicated in black in Figs. S10-S24) as the distance at which the ice-flow speed has declined to the $e$-th fraction of the respective centerline value. This method of detecting the glacier's lateral shear margins facilitates the localization of its lateral boundary especially if the outlet is rather weakly confined by the bed topography (e.g., Thwaites Glacier, Fig. S10). For outlets that are instead sharply confined by the bed (e.g., Pine Island Glacier) the velocity-deduced shear margins agree well with the lateral topographic confinement (Fig. S10). The values for $W$ are obtained along the first 20 km upstream of $p_1$. This captures the non-local influence of the confinement width $W$ (due to the non-local nature of the stress balance, Eq. 1) over the same distance upstream of $p_1$ for each outlet, independently of the length of its retrograde slope section. Over the same range the ice softness $A$ is averaged for each streamline. The above approach yields a set of measurements for each characteristic scale, spanning the observational range of this scale. The number of measurements for the scales $D, L$ and $A$ is given by the number of the glacier-specific (retrograde) streamlines, whereas for $W$ it is given by the 1-km sampling along the central streamline.

To ensure similarity between the measured outlets and comparability of their timescales via Eq. (4) we apply the following three criteria:

1. The outlets have to show **ice-stream characteristics**: the maximum flow speeds of the grounded part of the glacier have to be at least 100 m/yr (most of them are streaming at >500 m/yr).

2. **MISI susceptibility**: within the first 20 km upstream of the grounding line the calculated flux-averaged bed profile has to be below sea level and it has to continuously slope down in landward direction over a distance of at least 5 km (the vast majority of the analyzed glaciers has an initial retrograde bed section of 10 km or more).

3. **General geometric similarity**: while each glacier's profile has of course its own characteristics we require that fluctuations in the bed-slope direction have to be modest, i.e., we discard glaciers with more than one pronounced peak in the bed profile within the first 20 km (which excludes the strongly undulating Kohler Glacier tributaries from our analysis).

Applying these criteria, we obtain an ensemble of 13 outlet glaciers around Antarctica that are suitable to be involved in our analysis. Note that since the choice of the similarity criteria is not unique a different set of criteria than the one used here may lead to a different ensemble of outlets.

We choose Pine Island Glacier as the reference system for calculating the scaling ratios $\alpha$, $\beta$, $\delta$ and $\omega$ that enter the time-scaling relation. That is, the scales of all outlets analyzed in this study are expressed relatively to the scales of this reference. Pine Island Glacier is one of the most prominent Antarctic outlets and it is relatively well observed. Due to the nature of the conducted scaling analysis the scales calculated here could also be expressed relative to any other of the examined outlets without changing the results. Each scaling ratio of a specific glacier is considered as a distribution of results which accounts for the uncertainty of the two involved scales. That is, for each scaling ratio all measurements of the scale of the specific glacier and the reference are combined with each other. The distribution for the time scaling ratio $\tau$ is then calculated according to Eq. (4), based on the likely range (range between the $17^{\text{th}}$ and $83^{\text{th}}$ percentiles) of the distributions of $\alpha$, $\beta$, $\delta$ and $\omega$ by conducting Monte-Carlo simulations with a sample size of 1000 (Fig. 8). This likely range helps to confine the resulting time-scaling distributions: outliers in the measurements of the characteristic scales can generate strong uncertainties in the calculated scaling ratios which, in turn, are amplified through the non-linearity of Eq. (4) in combination with the Monte-Carlo method. Due to the large uncertainty in the observed along-flow bed shape (Morlighem et al., 2020) and the associated relatively large spread in the length scale, the $\delta$ scaling ratio is exempt from the Monte Carlo method and the median of the distribution is used in Eq. (4) instead.

## D2   Underlying datasets

The characteristic scales of the outlet glaciers are all obtained from datasets that represent present-day conditions of the Antarctic Ice Sheet. The bed topography stems from the Bedmachine Antarctica dataset (Morlighem et al., 2020) from which we infer the outlet-specific geometric scales $D$, $L$ and $W$. The regularly updated, continent-wide compilation involves data from various sources, including satellite, airborn radar, over-snow radar and seismic-sounding measurements. The novel dataset applies a mass conservation method to overcome limitations of the earlier BEDMAP2 compilation (Fretwell et al., 2013), revealing previously unkown bed features of Antarctica's outlets with substantial implications regarding their susceptibility to MISI.

Due to the absence of observed Antarctic ice-softness data, we use results from a present-day, continental-scale simulation of the Antarctic Ice Sheet, conducted with PISM in the course of the recent ISMIP 6 Antarctica intercomparison project (Seroussi et al., 2020). The thermo-mechanically coupled model is run into equilibrium under present-day conditions at a horizontal resolution of 8 km and a vertical resolution that ranges from 13 m at the ice base to 100 m at the top of the numerical domain. As part of solving the stress balance of the ice, PISM computes a temperature-dependent ice softness field, from which the vertically averaged form serves as the underlying dataset for our inference of the ice-softness scale $A$ (Fig. S9).

The ice flow field used to track the streamlines of the individual outlet glaciers is taken from Mouginot et al. (2019). It is the most recent and precise dataset of the present-day Antarctic ice surface velocity, derived by applying a combination of interferometric phase-mapping and speckle-tracking techniques to satellite measurements from the last 25 years.

*Author contributions.* Both authors designed the study, developed the theory and wrote the paper. JF conducted the model simulations and prepared the figures.

*Competing interests.* The authors declare that they have no competing interests.

*Acknowledgements.* This work was supported by the Deutsche Forschungsgemeinschaft (DFG) in the framework of the priority programme "Antarctic Research with comparative investigations in Arctic ice areas" through grants LE 1448/8-1 and WI 4556/6-1. We acknowledge the European Regional Development Fund (ERDF), the German Federal Ministry of Education and Research, and the Land Brandenburg for supporting this project by providing resources on the high-performance computer system at the Potsdam Institute for Climate Impact Research. Development of PISM is supported by NSF grants PLR-1644277 and PLR-1914668 and NASA grants NNX17AG65G and 20-CRYO2020-0052. We thank Camilla Schelpe and one anonymous referee for their valuable comments and suggestions which helped to improve the manuscript.

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

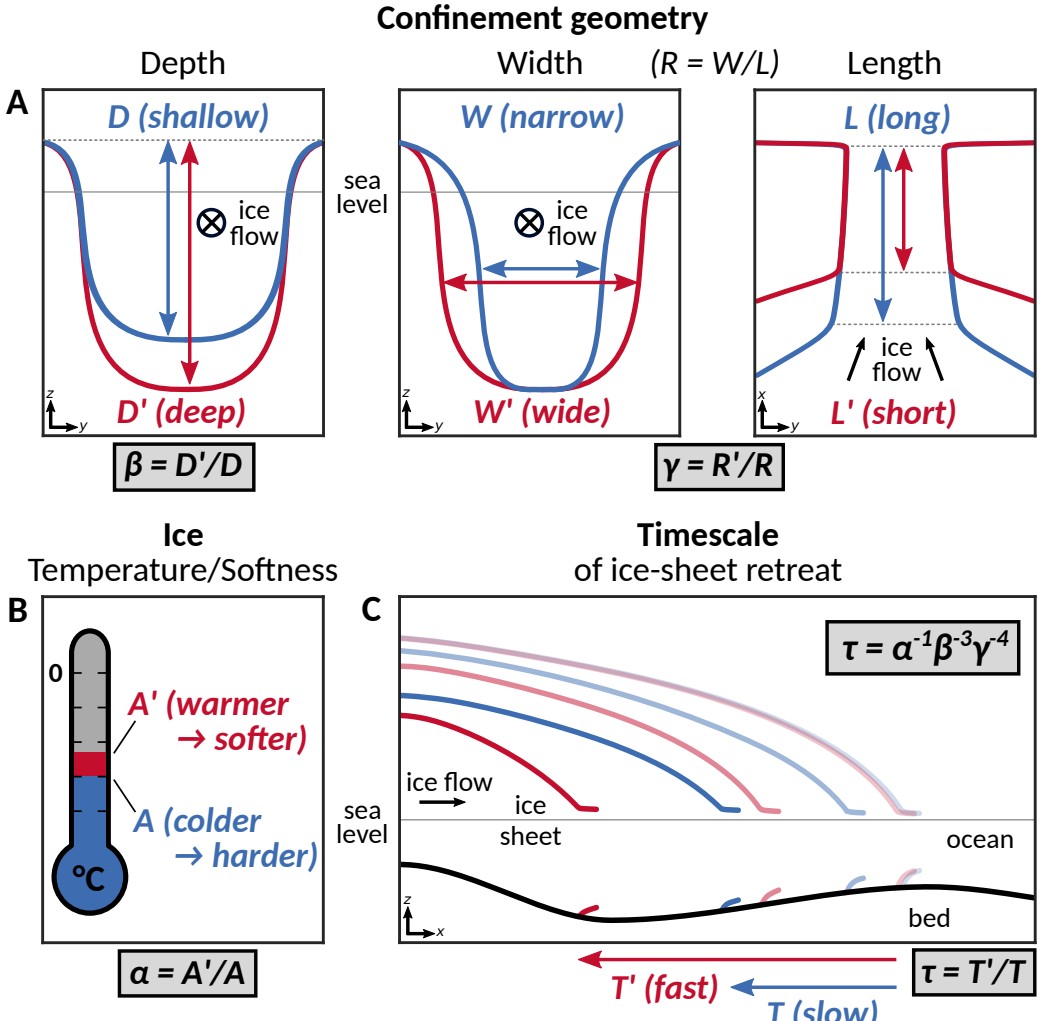

**Figure 1.** Schematic showing the influence of **(A)** the geometry (depth, width, length scales) of the confinement of fast ice flow, and **(B)** the temperature-dependent ice softness on **(C)** the timescale of outlet-glacier retreat. The retreat timescale is visualized in panel C by simultaneous snapshots of two similar outlet glaciers which share the same initial position (most translucent, oceanward profiles) and undergo unstable retreat down a retrograde slope (increasing opacity of profiles). Since the glaciers differ in one or more of the scales shown in panels A and B they retreat on different timescales: a deeper, wider and shorter confinement as well as warmer/softer ice lead to a faster retreat timescale (red contours in all panels). Conversely, shallower, narrower and longer confinements as well as colder/harder ice cause a slower retreat timescale (blue contours in all panels). The involved scaling variables and the scaling relation resulting from similitude analysis (Eq. 4) are given in grey boxes. The main flow direction is in $x$ direction. Note that the application of the scaling relation to the timescale of glacier *retreat* is based on the assumption that it is proportional to the timescale of glacier *flow* (see Sect. 3.2 and Appendix A2). For simplicity this schematic shows the case of a strongly topographically controlled ice-stream confinement.

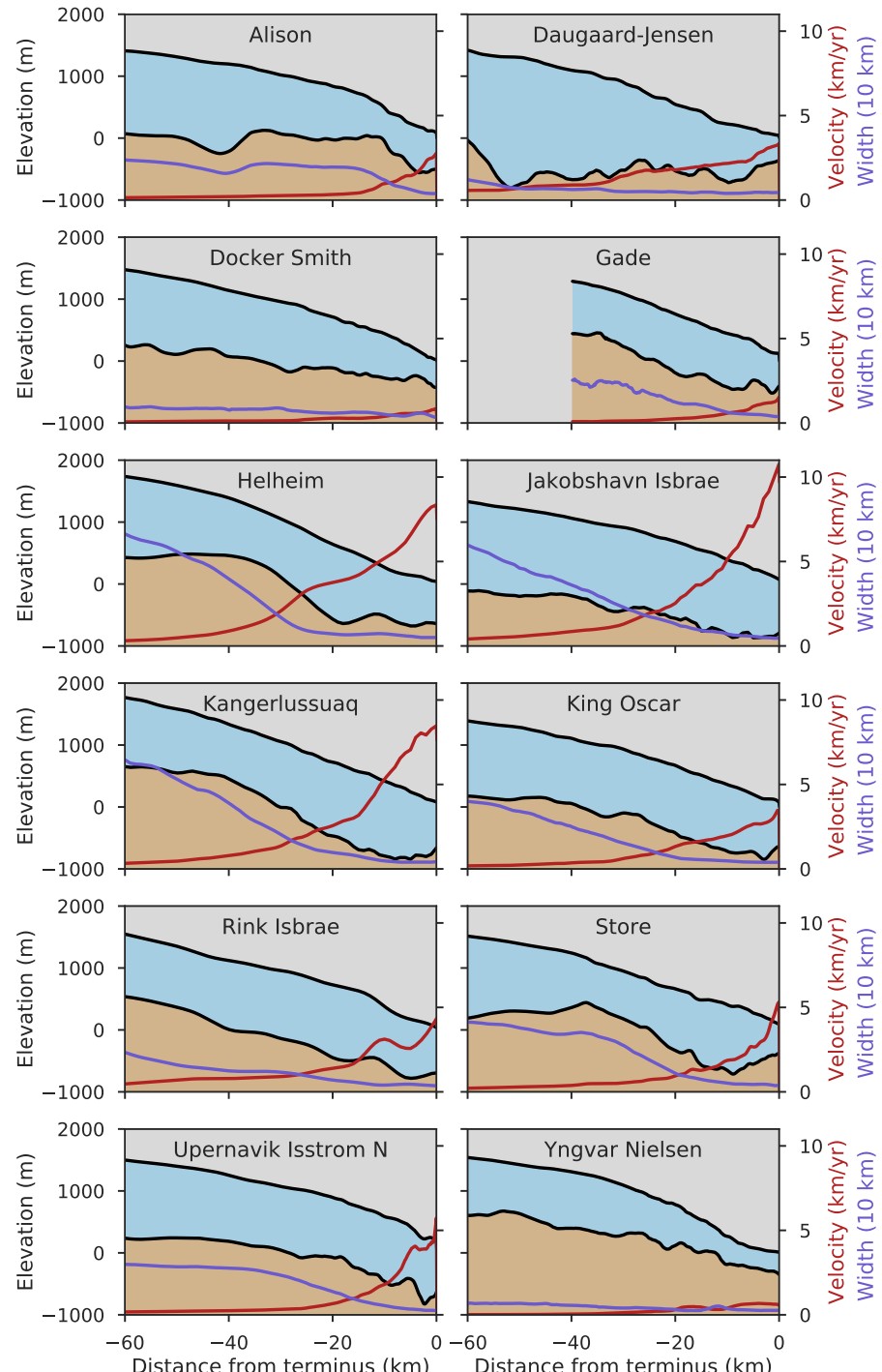

**Figure 2.** Profiles for twelve different glacier outlets across Greenland obtained from Beckmann et al. (2019), based on observational data. Bed topography in brown, glacier ice in blue. Ice speed and confinement width in red and purple, respectively ($y$-axis on right-hand-side).

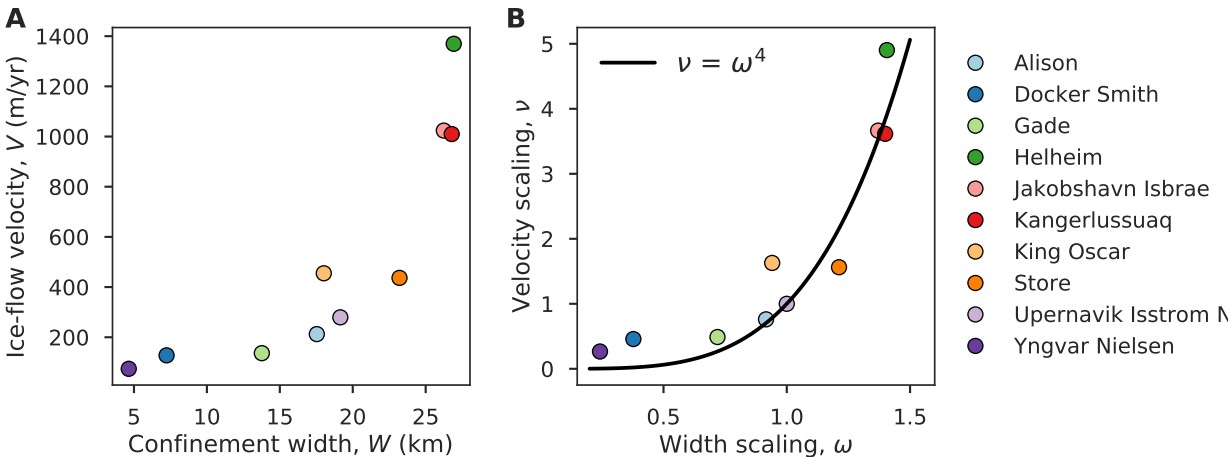

**Figure 3.** Scaling of observed ice-flow speed of Greenland outlet glaciers dependent on their width. **(A)** Ice-flow velocity inside the confinement versus confinement width extracted from observations. **(B)** Data normed to reference glacier Upernavik Isstrom N (light purple). The black curve shows the quartic relation between the scaling of the velocity, $\nu$, and the glacier width, $\omega$, as predicted by the scaling law (Eq. 4). Please note that there is no fudge factor between observation and theory in panel **B**. Glacier names are given in the legend.

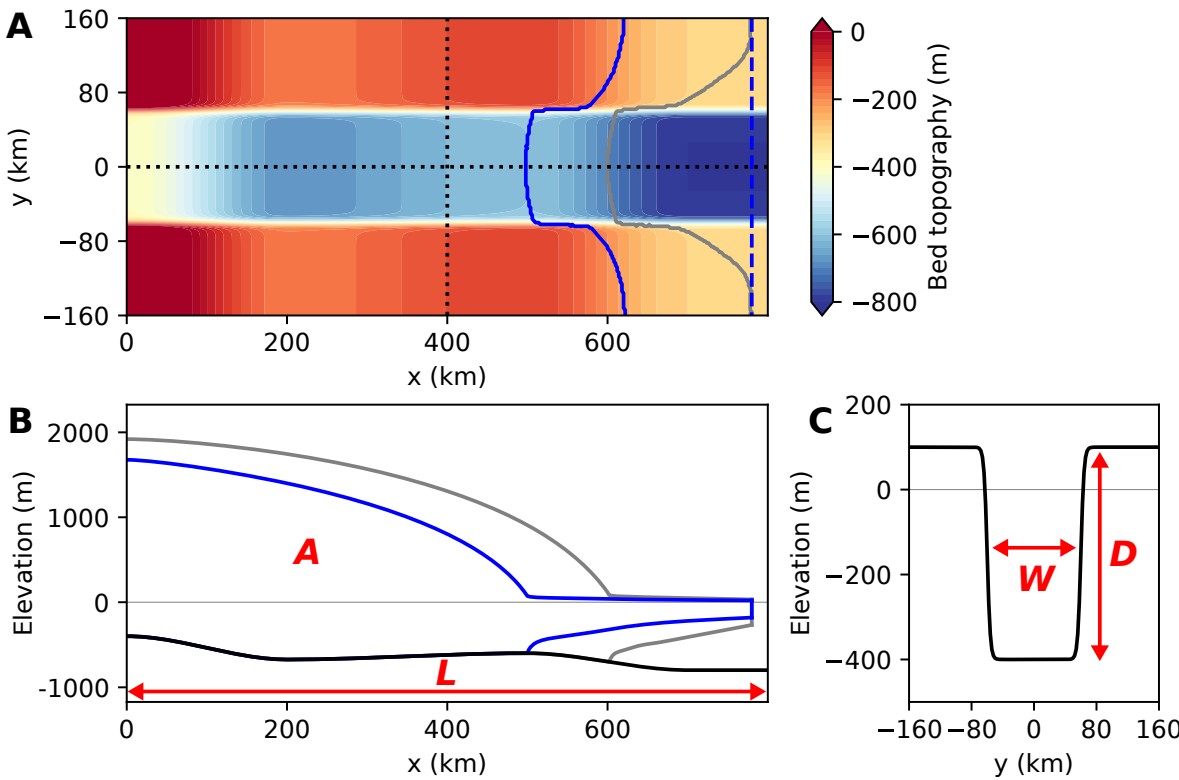

**Figure 4. (A)** Bed topography applied in the simulations (colorbar) with contours of the grounding line after $1,000$ model years (grey) and after equilibration (blue) for a stable configuration (bed-trough width $W = 100$ km). Fixed calving-front position shown by dashed line. Dotted lines denote cross sections **(B)** along the centerline of the setup ($y = 0$) and **(C)** across the bed trough ($x = 400$ km). The profiles of the ice-sheet-shelf system are shown in colors corresponding to panel **(A)**. The bed topography is shown in black. The length scale $L$, the width $W$ and the depth $D$ of the confinement, and the ice softness $A$ (property of the ice body), which are the parameters varied throughout the experiments, are highlighted in red.

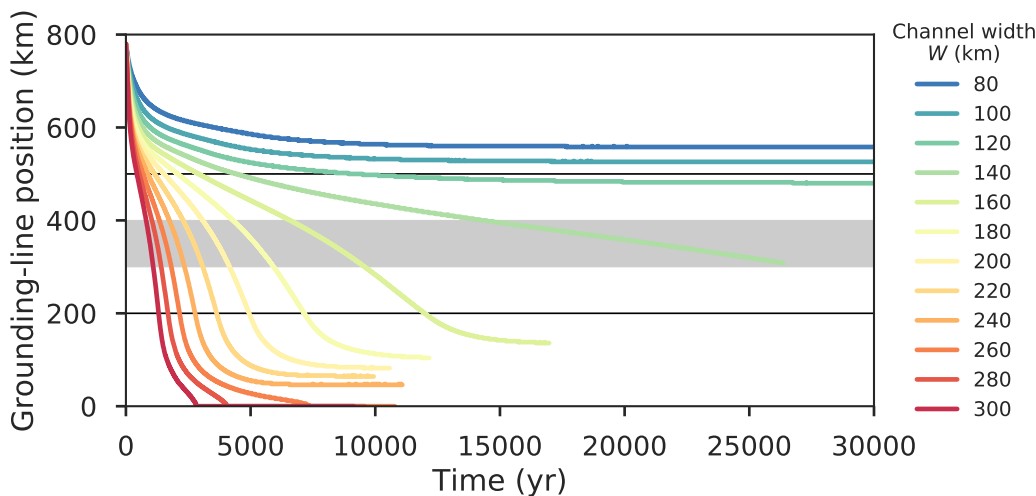

**Figure 5.** Evolution of the centerline grounding-line position (along $y = 0$) of the simulated ice-sheet-shelf system from in initialization to equilibration, here exemplarly shown for a variation of the channel width $W$. Horizontal black lines denote endpoints of the retrograde section of the bed topography (see Fig. 4B). The characteristic timescale is obtained by averaging the retreat time over the central 100-km long retrograde bed section (grey-shaded region).

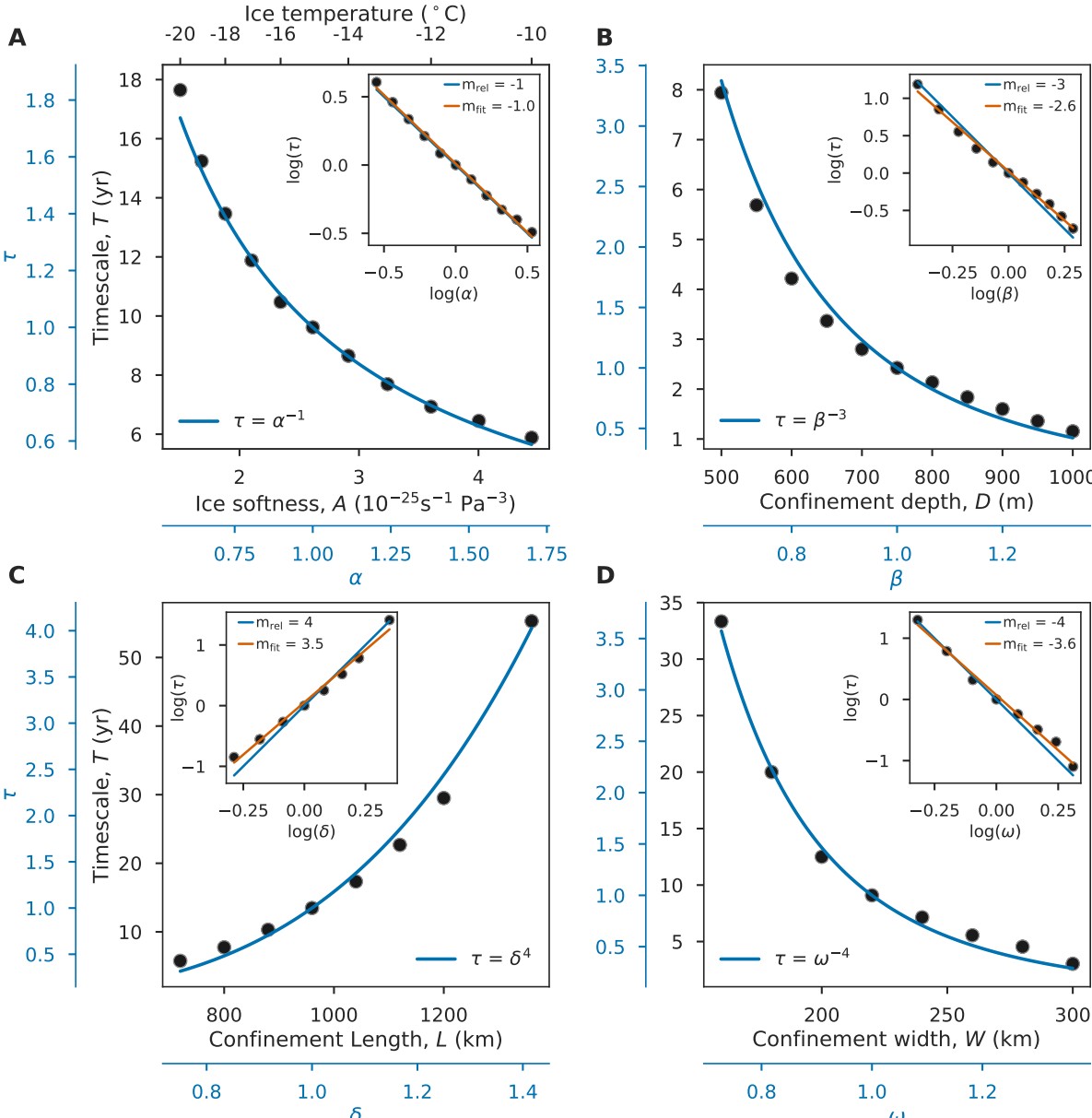

**Figure 6.** Time scaling of simulated unstable ice-sheet retreat. Characteristic retreat timescale $T$ down the retrograde bed slope (measured in years per km of retreat) dependent on **(A)** the ice softness $A$, **(B)** the depth of the confinement $D$, **(C)** the length scale of the confinement $L$ and **(D)** the width of the confinement $W$. The blue axes show the scaling if data is normed to a reference softness / confinement depth / confinement length scale / confinement width and the associated reference timescale. The blue curves give the scaling behavior according to the analytically-derived scaling relation (Eq. 4). The insets show double-logarithmic plots of the scaling ratios, with the slope of the approximately linearly increasing data points yielding the respective scaling exponent. The blue slope states the exponent expected from the theoretical scaling relation ($m_{\mathrm{rel}}$) and the orange slope gives the fit from linear regression of the simulation data ($m_{\mathrm{fit}}$). Please note that there is no fudge factor between simulations and theory.

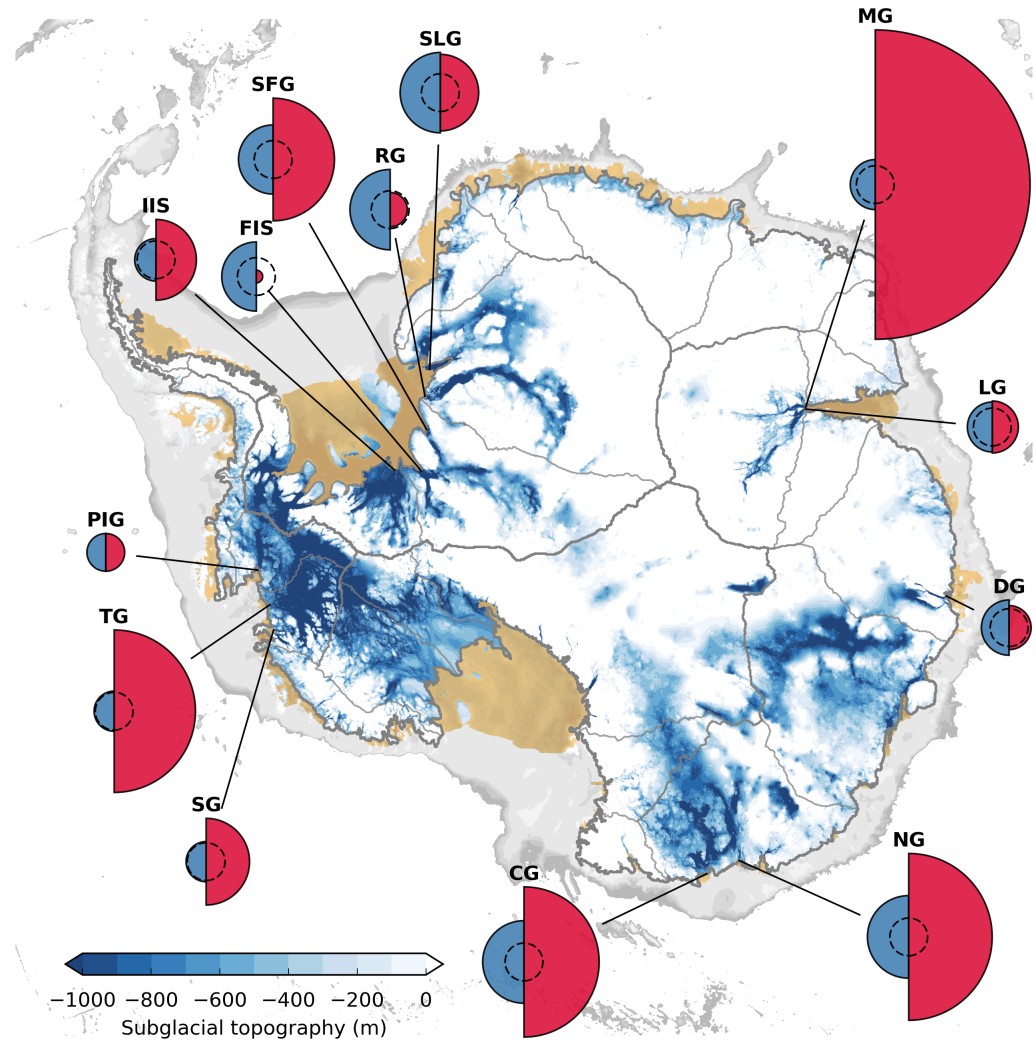

**Figure 7.** Inverse timescales of MISI-prone Antarctic outlet glaciers and their sea-level potential. Areas of red semi circles represent median values of the initial retreat speed after a potential destabilization ($<\tau>^{-1}$). Areas of blue semi circles correspond to the sea-level equivalent of the ice in the marine portions of the drainage basin(s) of the respective outlets (SLE). All values are calculated relative to the reference system Pine Island Glacier (PIG, dashed circles), with reference values $<\tau_{\mathrm{PIG}}>^{-1} = 1$ and $\mathrm{SLE}_{\mathrm{PIG}} = 0.53$ m. Timescale values and abbreviations of the outlet names are given in Table 3.

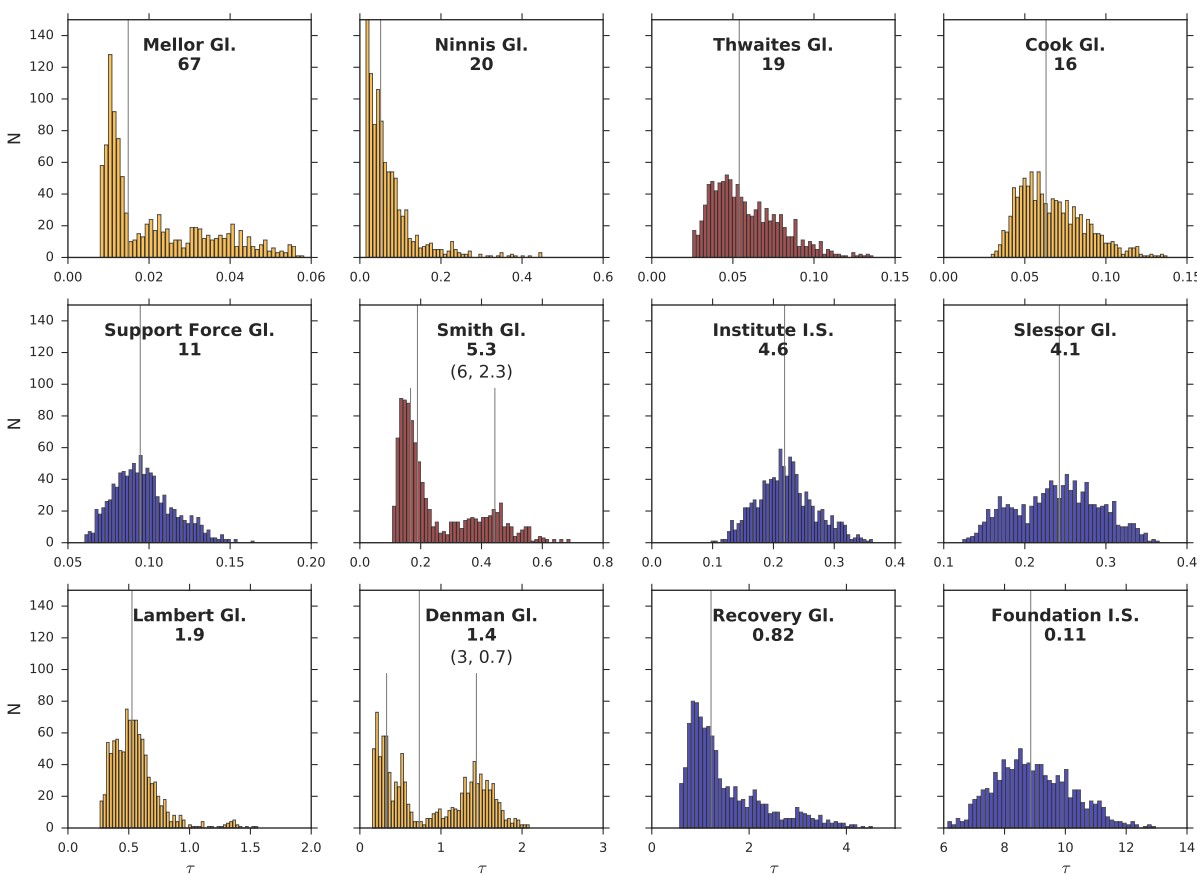

**Figure 8.** Probability distributions of the retreat-timescale ratios. Median values $<\tau>$ are shown by vertical lines. The values given below the glacier names are the inverse of the median $(<\tau>^{-1})$, i.e., the retreat-speed ratios. Locations of the outlets: Amundsen Sea Sector (red), East Antarctica (yellow) and Filchner-Ronne Ice Shelf (blue).

**Table 1.** Definition of scales and their ratios.

| Category | Scaling ratio | Letter | Definition |
|---|---|---|---|
| Thermodynamics | Softness | $\alpha$ | $A'/A$ |
| Geometry | Depth | $\beta$ | $D'/D$ |
| | Length | $\delta$ | $L'/L$ |
| | Width | $\omega$ | $W'/W$ |
| | Depth gradient | $\Gamma$ | $\beta/\delta = (D'/L')/(D/L)$ |
| | Horizontal aspect ratio | $\gamma$ | $\omega/\delta = R'/R = (W'/L')/(W/L)$ |
| Kinematics | Time | $\tau$ | $T'/T$ |
| | Velocity | $\nu$ | $\delta/\tau = V'/V = (L'/T')/(L/T)$ |
| | Volume flux | $\mu$ | $\nu\,\beta\,\omega = (V'\,D'\,W')/(V\,D\,W)$ |

**Table 2.** Physical constants and parameter values as prescribed in the simulations.

| Parameter | Value | Unit | Physical meaning |
|---|---|---|---|
| $a$ | 0.15 | m yr$^{-1}$ | Surface mass balance |
| $A_0$ | $3.615 \cdot 10^{-13}$ | Pa$^{-3}$ s$^{-1}$ | Pre-exponential factor in Eq. (C1) |
| $C$ | $3.981 \cdot 10^{6}$ | Pa m$^{-1/3}$ s$^{1/3}$ | Basal friction parameter in Eq. (C2) |
| $g$ | 9.81 | m s$^{-2}$ | Gravitational acceleration |
| $m$ | 1/3 | | Basal friction exponent in Eq. (C2) |
| $n$ | 3 | | Exponent in Glen's law |
| $s$ | $2.5 \cdot 10^{-4}$ | | Reference mean slope magnitude of retrograde bed section |
| $Q$ | $6 \cdot 10^{4}$ | J mol$^{-1}$ | Activation energy in Eq. (C1) |
| $R_g$ | 8.314 | J K$^{-1}$ mol$^{-1}$ | Universal gas constant in Eq. (C1) |
| $T_{\mathrm{ice}}$ | $\{-20, -19, ..., -11, -10\}$ | $^{\circ}$C | Ice temperature entering Eq. (C1) |
| $d_c$ | $\{500, 550, ..., 950, 1000\}$ | m | Depth of bed trough compared with side walls, |
| | | | entering Eq. (1) of Cornford et al. (2020) |
| $f_c$ | 4 | km | Characteristic width of bed-trough side walls, |
| | | | entering Eq. (1) of Cornford et al. (2020) |
| $l_c$ | $\{720, 800, ..., 1280, 1360\}$ | km | Length of bed trough |
| $w_c$ | $\{40, 50, ..., 140, 150\}$ | km | Half-width of bed trough, |
| | | | entering Eq. (1) of Cornford et al. (2020) |
| $x_{\mathrm{cf}}$ | $0.975 \cdot l_c$ | km | Position of fixed calving front |
| $\rho$ | 918 | kg m$^{-3}$ | Ice density |

**Table 3.** Median values of inferred scaling-ratio distributions for ice softness $<\alpha>$, confinement depth $<\beta>$, confinement length $<\delta>$, confinement width $<\omega>$ and time $<\tau>$. Last column gives the likely range of $\tau$ (17th - 83rd percentile).

| Outlet glacier | $<\alpha>$ | $<\beta>$ | $<\delta>$ | $<\omega>$ | $<\tau>$ | $\Delta\tau$ |
|---|---|---|---|---|---|---|
| **Amundsen Sea Sector** | | | | | | |
| Pine Island Glacier (PIG) | 1.0 | 1.0 | 1.0 | 1.0 | 1.0 | - |
| Thwaites Glacier (TG) | 0.77 | 0.63 | 0.58 | 2.08 | 0.054 | 0.025 - 0.137 |
| Smith Glacier (SG) | 1.13 | 0.91 | 0.29 | 0.53 | 0.189 | 0.107 - 0.692 |
| **East Antarctica** | | | | | | |
| Cook Glacier (CG) | 0.76 | 0.53 | 1.11 | 3.76 | 0.063 | 0.029 - 0.138 |
| Ninnis Glacier (NG) | 0.42 | 1.04 | 0.43 | 1.05 | 0.051 | 0.014 - 0.449 |
| Denman Glacier (DG) | 0.56 | 0.81 | 0.28 | 0.45 | 0.733 | 0.153 - 2.091 |
| Lambert Glacier (LG) | 0.48 | 0.82 | 0.55 | 0.9 | 0.526 | 0.262 - 1.562 |
| Mellor Glacier (MG) | 0.48 | 1.8 | 0.18 | 0.45 | 0.015 | 0.008 - 0.058 |
| **Filchner-Ronne Ice Shelf** | | | | | | |
| Slessor Glacier (SLG) | 0.65 | 0.86 | 1.27 | 2.18 | 0.243 | 0.123 - 0.366 |
| Recovery Glacier (RG) | 0.54 | 1.2 | 1.05 | 0.9 | 1.217 | 0.576 - 4.537 |
| Support Force Glacier (SFG) | 0.62 | 1.12 | 0.49 | 0.98 | 0.095 | 0.060 - 0.165 |
| Foundation Ice Stream (FIS) | 0.45 | 1.81 | 2.46 | 1.13 | 8.861 | 6.127 - 12.968 |
| Institute Ice Stream (IIS) | 0.6 | 0.94 | 1.55 | 2.26 | 0.218 | 0.097 - 0.363 |