# Peer review of "Timescales of outlet-glacier flow with negligible basal friction: Theory, observations and modeling"

_The Cryosphere, 2022_

## Author Response (AR1)

Dear Editor,

We would like to thank you for the careful handling of the review process. We are grateful for the valuable comments and suggestions from both Referees as they really helped to improve our manuscript. In our revised manuscript we addressed all the issues raised by the Referees (see our point-to-point answers below). Please find below the *Referees' comments in italics* and our detailed response in blue.

Best wishes,
J. Feldmann et al.

**RC1: 'Comment on tc-2022-141', Anonymous Referee #1**

*This paper uses the similarity principle applied to the SSA neglecting the basal friction with the goal to infer the time scale of marine ice stream retreats in Greenland and Antarctica. The paper considers idealized setup along with specifically chosen glaciers for Greenland and Antarctica and shows relatively good agreements with the theory.*

*I found the paper really well written and organized. I would like to thank the authors to have spent the time on both aspects. I enjoyed reading this manuscript. While working with 3-D complex models is becoming more of a norm, it is refreshing to see that simple mathematical arguments (derived from equations used in the 3-D models) remain useful to understand or give a sense on dynamical principals. While the manuscript is very mathematically driven, the authors carefully left the details in the appendix while leaving the basic understandings of the method in the main text. Doing so results in a relatively short manuscript with little dilution from the mathematical details and well suited for a quick read.*

*I would fully support the publication of this publication after minor revision. Main and minor comments are following.*

We would like to thank Referee#1 for their willingness to review our manuscript, the helpful comments and the constructive suggestions. We are glad for the Referee's positive assessment of our study and are happy to hear that they would support the publication in TC. We addressed all the points raised by the Referee (see our detailed response below) which helped to improve our manuscript.

**Main Comments**

1. *In comparing your similarity principle to Greenland glaciers, you set aside a couple of the glaciers and argued that physical properties of the glaciers break the similarity assumption. Did you check that it was simply the case that these glaciers behaved with respect to a similarity principal for which the main balance in the momentum equation is between the basal friction and the driving stress?*

   In fact we cannot exclude that for the discarded Greenland glaciers a different approximation of the stress balance might be valid. However, to tackle this question, knowledge on the basal conditions of all examined outlets would be required, i.e., their basal stress field. We anticipate that obtaining such data from numerical modeling would be a comparatively large effort and introduce additional uncertainty to the analysis. Though an in-depth analysis of the prevailing stress regimes of Greenland's outlet glaciers would surely be of great insight, we deem such an examination to be beyond the scope of our study.

2. *Since you choose to neglect the basal friction term in your similarity discretization, why did you choose a Weertman-type sliding law as opposed to a Coulomb-derived sliding law (Schoof (2005), Leguy et al. (2014), Tsai et al. (2015) ) in your idealized setup? A Coulomb*

*sliding law has the advantage to go to zero at the groundline line by design. Please discuss this point in your manuscript.*

We see the reviewers point here since it is desirable to have a basal stress field that drops to zero when approaching the grounding line, especially in the context of our study. For an application of a Coulomb-limited law to our setup, this transition zone of basal resistance, however, is a rather narrow region directly upstream of the grounding line. Among other model realizations, we used the Coulomb-limited law by Tsai et al. (2015) in the course of the MISMIP+ intercomparison exercise, which is based on a very similar setup as the one used in the present study. The results showed that the Coulomb-limited law changed the basal friction field only in a 2-km narrow stripe at the grounding line.

As in earlier idealized simulations, in the present study we apply an interpolation of the basal friction at the grounding line between the ice shelf (zero basal friction) and the upstream grounded ice (non-zero basal friction) which - on the used resolution of 2 km - has a similar effect of smoothing the basal friction close to the grounding line as using the Coulomb-limited law. In fact, a comparison of the MISMIP+ results of both model realizations showed that both realizations led to qualitatively the same response to the applied perturbations (with some deviations in the initial grounding-line position and the response magnitude). Due to these facts and since the combination of Weertman-type sliding plus basal friction interpolation used in the present study proved to be appropriate in several previous idealized modeling studies we applied it also in the present study. As suggested by the Referee we added a discussion of the above points to the manuscript (P18, L521-531).

3. *How can you compare your Antarctic theoretical computation with actual retreat rates? Could you compare them using the data from Rignot et al. 2019? You have found an elegant way in obtaining a retreat rate with a simple method, it would be super useful to show that the order of magnitude matches current observation in a way.*

We fully agree that comparing the ratios calculated in our study with observations in Antarctica would be very interesting. The effort to do so would however be substantial because we would have to define a timescale for each of the observed glaciers, which is possible but not trivial, and translate these absolute numbers to ratios which would require a proper uncertainty analysis, especially because dividing two uncertain numbers can potentially generate strong uncertainties. If the Referee and the Editor agree we would very much like to defer the task to a future study that stands by itself.

4. *While I understand that textbooks are a rich resource of information, using them for citations should be a last resort and when doing so, please mention the chapter and section or page number (Cuffey and Paterson 2010) is a rather large volume! You have used book references very often while papers would have been more appropriate. I will try to address a few of them below.*

We thank the Referee for the advice. We removed all textbook citations and added the

paper suggestions given by the Referee below under "Minor comments".

**Figures**

*Figure 1, panel C: Please describe the meaning of each color shade. Also, are you using different values of A in order to obtain each profile starting from the most advanced profile? (I thought this was not very clear in the text or caption.)*

As suggested by the Referee, we now give a more detailed explanation of panel C in the figure caption regarding the plotted glacier profiles.

*Figure 4, panel B: I don't understand what the ice softness indication "A" is doing on the figure. Please clarify or remove it.*

The ice softness $A$ is one of the four scaling parameters varied in our simulations as stated in the figure caption. The other three parameters, $L$, $W$ and D do all concern spatial dimensions and are thus equipped with a double-headed arrow. Since the ice softness $A$ is a property of the ice body itself we placed the letter inside the ice body. We added a brief remark on that to the figure caption.

*Figure 7: please give a reference value for the blue and red semi-circle for your reference glacier (PIG).*

The reference values for PIG are now provided in the figure caption. While checking for the SLE reference value we noticed that in the original version of the figure we accidentally used a reference value of 1 m instead of 0.53 m for calculating the SLE ratios. We corrected this and thus the areas of the blue semi circles in Fig. 7 are now larger by a factor of about two. Note that the correction of this mistake does only concern the plotted SLE ratios but no other calculations in this study.

**Minor comments:**

*Page 4, line 93: replace the citation "Greeve and Blatter, 2009" by Glen (1955).*

Done.

*Page 4, line107: The citations to reference the "Observations and laboratory studies" are wrong. Schoof 2007 and Haseloff 2015 use the value in their modeling effort. Instead, cite Duval (2013). Also, n cannot be observed directly as it is a parameter in a law which best fit data. Instead, I would replace "Observations and laboratory studies" by simply "Laboratory studies".*

We thank the Referee for this valuable hint! We modified the text and replaced the citations according to the Referee's suggestions (P4, L112-113).

*Page 4 line 114: see remark above regarding Cuffey and Paterson.*

Citation replaced by Glen (1955).

*Page 5 line 120: you can also cite Leguy (2015, chapter 7.1.5) who derived a relation for buttressing that is inversely proportional to the width of the bed. (Note that I do not expect people to read the dissertation of an author, I just happen to know his work having collaborated with him.)*

We thank the Referee for this hint and included the citation.

*Page 6, line 167: what did you modify in the MISMIP+ setup. Please add the relation you use in your bed topography derivation (in the appendix).*

We added a detailed description of the modified bed topography to the Appendix (P17, L495-506), as suggested by the Referee.

*Page 8, line 216: Please remove the part of the sentence "might have been destabilized by recent oceanic warming."  It is highly speculative hence unnecessary here unless you can support the claim.*

Done.

*Page 10, line 278-279: for the reason you mention here, why not running the model using a Coulomb friction law?*

Please see our response to the Referee's related comment above (main comment #2).

*Page 10, line 279: Replace the citation by (Schoof (2005), Martin et al. (2011), Leguy et al. (2014))*

Done.

*Page 11-12, code and data availability: The code, data and simulation setup should be available prior to the publication of the manuscript. Make sure that it is the case and replace the wording "will be" by "is". At this point I could access your data.*

The used model code and the simulation data are now stored in zenodo archives. In the code and data availability statement of the manuscript we now cite these publications together with their DOIs (P12, L350-352).

*Page 12, line 358: citation, see remark above.*

Citation replaced (P12, L354-356).

*Page 12, line 365: I don't see the necessity in introducing the scalings Sx and Sy. Why not directly using L and W?*

The idea behind starting with *Sx* and *Sy* is to keep the beginning of the derivation as general as possible. That is, we apply the specific case of laterally confined flow – which we place into the coordinate system such that *L* = *Sx* and *L* = *Sy* – only after the non-dimensionalization. This might be a matter of taste and we would like to keep it this way. However, if both the Referee and the Editor are convinced that introducing *Sx* and *Sy* is very unnecessary, we would be willing to change the formulation.

*Page 12, line 366: Please, introduce the scaling x\*=x/L and y\*=y/W for clarity.*

Done (P13, L376)

*Page 13, line 381: Please define the O symbol as the Landau notation for non-mathematicians. And technically speaking, the leading order in Eq.(A3) is O(1/R) (if we account for the square root).*

Done (and thanks for the correction) (P14, L412-413).

*Page 14, line 399: citations, replace with Glen (1955) and Duval (2013).*

Done.

*Page 14, line 412: You argue that the velocity shape of the Rink Isbrae glacier near the terminus is ground to discard this glacier from your analysis. Why then keep Upernavik Isstrom N which exhibit a similar pattern?*

Thanks for pointing out this important detail. We added a brief statement on why Rink is discarded and why Upernavik not (P16, L466-469).

*Page 15, line 442: Please give the uniform rate of SMB you used in your simulations.*

Done (P18, L512).

*Page 15, line 444: replace citation with Glen (1955)*

Done.

*Page 15, line 449: Add chapter to citation.*

We replaced the textbook citation with the paper of Pattyn et. al, 2013 which uses the sliding law in the same form as we do (P18, L518).

*References:*

*Rignot, E., Mouginot, J., Scheuchl, B., Van Den Broeke, M., Van Wessem, M. J., & Morlighem, M. (2019). Four decades of Antarctic Ice Sheet mass balance from 1979–2017. Proceedings of the National Academy of Sciences, 116(4), 1095-1103.*

*Glen, J. W. (1955). The creep of polycrystalline ice. Proceedings of the Royal Society of London. Series A. Mathematical and Physical Sciences, 228(1175), 519-538.*

*Duval, P. (2013). Creep behavior of ice in polar ice sheets. In The Science of Solar System Ices (pp. 227-251). Springer, New York, NY.*

*Leguy, Gunter. The effect of a basal-friction parameterization on grounding-line dynamics in ice-sheet models. New Mexico Institute of Mining and Technology, 2015.*

*Schoof, C. (2005). The effect of cavitation on glacier sliding. Proceedings of the Royal Society A: Mathematical, Physical and Engineering Sciences, 461(2055), 609-627.*

*Martin, M. A., Winkelmann, R., Haseloff, M., Albrecht, T., Bueler, E., Khroulev, C., and Levermann, A.: The Potsdam Parallel Ice Sheet Model (PISM-PIK) – Part 2: Dynamic equilibrium simulation of the Antarctic ice sheet, The Cryosphere, 5, 727–740, doi:10.5194/tc-5-727-2011, 2011. 6*

*Leguy, G. R., Asay-Davis, X. S., & Lipscomb, W. H. (2014). Parameterization of basal friction near grounding lines in a one-dimensional ice sheet model. The Cryosphere, 8(4), 1239-1259.*

*Tsai, V. C., Stewart, A. L., Thompson, A. F.: Marine ice-sheet profiles and stability under Coulomb basal conditions, J. Glaciol., 61, 205–215, doi:10.3189/2015joG14j221, 2015. 28, 76, 77*

**RC2: 'Comment on tc-2022-141', Camilla Schelpe:**

*In this study, the authors derive a scaling relation which provides a simple prediction for the characteristic timescales of outlet glaciers in Greenland and the Antarctic. They assume the flow can be described by the SSA and neglect basal friction in the momentum conservation equations. Based on the width of these outlet glaciers being an order of magnitude less than the length of the flow, they determine the leading order terms in the momentum equations and show that the driving stress is balanced by the lateral shear stresses for these geometries. This leads to a dimensionless relationship which compares ice flows with similar properties. It is a clean and simple formulation which, by focussing on similitude, abstracts many of the complex interactions which govern the ice flow. The resulting scaling relation only requires the geometrical properties of the outlet glaciers (depth, width and length) together with ice softness, to determine the characteristic flow timescale. The derived relationship does not make absolute predictions, but instead makes predictions relative to other glaciers with similar properties.*

*The authors then go on to test this relationship thoroughly. First, they compare the predictions against the timescale inferred from velocity measurements of various Greenland outlet glaciers which exhibit similar topographic properties. These experimental results are promising. Second, they compare the predictions against the retreat timescale from idealised numerical simulations performed within the PISM ice sheet model. This comparison requires the assumption that the flow timescale and retreat timescale are proportional. In this idealised simulation that does indeed appear to hold true, and the authors find excellent agreement to their scaling relation. Finally, they use the derived scaling relationship to make predictions of the relative retreat timescale for a number of Antarctic outlet glaciers with a retrograde bed that may be prone to MISI-type retreat.*

*I enjoyed reading the paper. The manuscript was well laid out, with the complexity of the mathematical derivations, and technical details of data extraction and experimental set-up nicely compartmentalised into separate appendices. The results are well explained, with helpful interpretations of the intuition behind a number of the mathematical results. I also appreciated the comparison in Sec 5 to their earlier work (Levermann and Feldmann, 2019) which considered a 1-D flowline with basal shear stresses included, but lateral drag neglected. Taken together, the conclusions from these two studies can be considered to give a range of predictions under the differing assumptions.*

*I have some comments and questions about the study, but I'm hoping these can be addressed through adding a bit more discussion to the manuscript rather than requiring any major changes to the results. I would fully recommend publication with these additions.*

We are grateful for the willingness of Camilla Schelpe to review our manuscript and for the constructive remarks, questions and suggestions, which helped to improve our manuscript. We are pleased by the positive assessment of our manuscript and are delighted to read that the Referee recommends the publication of our study subject to minor revisions. We addressed all the Referee's comments in the revised version of the manuscript.

**Specific Comments:**

*This study neglects the contribution of basal shear stresses, which the authors justify through reference to various papers (L81-L83) which infer a low basal friction coefficient for the rapidly sliding ice stream outlets of Antarctica and Greenland. Since a low basal friction coefficient, if combined with rapidly flowing ice, doesn't necessarily translate to negligible basal shear stress, it would be good if the authors discussed this decision a bit more in the manuscript. Maybe they could discuss the expected dominance of lateral drag in deep, narrow confinement channels. And/ or the results from the idealised Antarctic simulation which includes basal friction, could be included as a post-hoc justification?*

We thank the Referee for this hint for improvement. As suggested, we added a statement to Section 2, in which we refer to the connection between basal friction and basal stresses as well as the role of deep, narrow bed troughs (P3, L79-88).

*It was only after reading the paper fully that I understood the distinction between the flow timescale which is inferred from the surface velocity of the ice stream and used in the derivation of the scaling relation; and the retreat timescale for the speed of grounding line retreat, which is that simulated and predicted for the Antarctic outlet glaciers. I think it would be helpful for the reader if that distinction was emphasised in the introductory section of the text, and added as a fourth point in the potential limitations listed in Sec 5.*

We would like to thank the Referee for highlighting that this was still not as clear as we wished it to be. As suggested by the Referee, we added a statement to the Introduction (P3, L71-72) and also a fourth point to the list of limitations in the discussion (P10, L301-307).

*Related to the above point, the excellent agreement between the timescale for grounding line retreat in the idealised Antarctic simulations, and that predicted theoretically by the flow timescale, is perhaps surprising. It suggests a mathematical relationship which holds true in this idealised set-up. The authors allude to this on L156: "Grounding line retreat depends on the divergence of grounding-line discharge, i.e., on the divergence of the flow speed at the grounding line. If we were to seek a relation of the grounding-line retreat, we could make the assumption that the retreat speed of an outlet glacier is proportional to its flow speed." Could the authors include the mathematical reasoning for this? This would also help explain under what conditions the assumption that "retreat timescale = flow timescale" is correct and thus how those conditions are being met for the retrograde slope in the idealised simulations. Related to this I think the commentary on L11. and in Fig 1 caption that, "the flow velocity and its spatial derivative are proportional" may be misleading. If I have worked out this relationship correctly, I believe it should be that $\partial h/\partial t \propto h \times \partial u/\partial x$ ? Not $u \propto \partial u/\partial x$ ?*

We are grateful that the Referee has pointed out this unsolved issue. We now provide a mathematical derivation in which we clearly state under which assumptions the scaling relation (Eq. 4) can be applied to both the time scaling of *flow* and *retreat*. For this purpose, we added a new section to the Appendix (A2 "Relation between timescales of ice flow and grounding-line

retreat" on page 14-15, lines 415-453). For the derivation we carry out a dimensional analysis of the ice thickness equation (continuity equation), which links changes in the ice thickness in the course of the retreat (associated with the retreat timescale) to the ice flux divergence (associated with the flow timescale). The main assumption in the derivation is that the velocity change due to the dynamical imbalance in the course of retreat is proportional to the equilibrium velocity. We now refer to this in the abstract (P1, L11-12), the discussion section (P10, L301-310) and in the revised paragraph mentioned above by the Referee (P6, L162-168).

*In the experimental testing of the predicted scaling relation to the grounded Greenland outlet glaciers, the authors take the average over 60km-0km upstream of the grounding line for the estimated width, and 60km-20km for the estimated velocity. I understand the authors' comment that the velocity is being cut off to avoid pollution from the ice-ocean interactions for the last 20km, but the velocities generally seem to increase in the last 20km coinciding with the width narrowing. Therefore, would it not make more sense to also exclude the last 20km from the width estimation so that you are comparing like-with-like? How sensitive is the fit of the data to these choices? The good fit of the scaling relation to the Greenland outlet glaciers lends confidence to the similitude approach being valid across real word glaciers, i,e. that glaciers exhibit enough similarities that this simple scaling can be applied across them. It therefore seems important to make sure the conclusions are robust, and not sensitive to these slightly arbitrary choices.*

This is an important point mentioned by the Referee. To get an idea of the sensitivity of our results to the averaging ranges we plotted the data for four choices for the averaging ranges of the velocity scale $V$ and the width scale $W$, resulting from the four possible combinations of (not) excluding the last 20 km of the velocity and/or width profiles. This includes

1. the case we used in the original submission which only takes into account the first two thirds of the velocity profile but the entire width profile (labeled $V2/3$, $W$full),
2. the case suggested by the Referee which leaves out the last third for both profiles (labeled $V2/3$, $W2/3$),
3. the case in for which in both profiles the last third is neglected (labeled $V2/3$, $W2/3$), and
4. the case in which the full range of both profiles is used (labeled $V$full, $W$full).

The results are given in the new Supplementary Fig. S2, showing that the deviations vary throughout the four different cases. In fact, the largest deviations from the theory occur if the full velocity profile is taken into account (panels C and D), which might be due to the possibility that the ice velocity at the glacier terminus is affected by ice-ocean interactions. We added a statement on the sensitivity of our results to the choice of the averaging range on P17, L482-486.

*For the plots in Figures 3, 6 and S4, would it make sense to use a log plot so that the predicted scaling relationship gives a straight line (with the gradient equal to the exponent in the scaling law)? Deviation from the expected behaviour would then perhaps be easier to see by eye. It would also be helpful to plot the OLS estimate from the data and compare the gradients.*

We very much appreciate this comment and in principle agree that a log-log plot might be helpful. However these kinds of data presentation only give reasonable results when they are carried out

over at least three orders of magnitude on the x-axis. For smaller ranges the applicability is reduced due to the unavoidable noise in the results. We find this to be the case for the observations and even see it in the numerical simulations. We have however added log-log plots to figure 6 in the main paper, (referred to on P7, L193-195) and believe that this is a very nice addition. Thanks again for that.

**Technical Corrections:**

*There are a number of places in the text which refer to the scaling relation being linear in the ice temperature: L8, L129, L253. However, my understanding is that the linear relationship being referred to is to the ice softness A. The ice softness is temperature dependent, but not linearly dependent on temperature, I believe?*

This is indeed true and we thank the Referee for discovering this mistake, which we corrected (P1, L8; P5, L135; P9, L261).

*L465-L468. I found this description of extracting the length scale confusing. Could it be rewritten? Is it just a justification for setting L=1/S in the scaling equations, or is something else going on here?*

Thanks for pointing out that this description might be confusing. Indeed, the we wanted to justify that we set *L=1/S* but apparently we missed the target. As requested by the Referee we rewrote this section, giving a  more concise and clearer statement on how the length scale is obtained (P19, L545-546).

*Is there a reason for picking p1 to p1+20km as the distance over which to estimate W? Naively I think I would have expected the estimate to be the average over p1 to p2.*

Yes. The reason for choosing the same distance for all outlets lies in the non-local nature of the stress balance that underlies the derived scaling relation. By choosing a range that goes farther inland than the actual retrograde bed section we capture the influence of the upstream confinement on the downstream ice portion, i.e., the grounding-line region, over the same distance for each outlet. We added a brief statement on this to the manuscript (P19, L552-554).

*Clarify that these multiple combinations are coming from the multiple flow lines for each outlet glacier. (Unless I have misunderstood, in which case even more clarification needed!)*

Yes, that's the case. Clarified (P19, L554-557).

*Could you elaborate on why you have chosen the 17th and 83rd percentiles, rather than using the 5th-95th percentile range?*

We added a brief statement on why we chose those percentiles (P20, L575-581).

*Delete "respectively".*

Done.

*This reads as if the uncertainty range in Table 3 reflects the breakdown in the similitude requirement. My understanding was that the uncertainty range still assumes geometric similarity and that the scaling relation holds; instead it reflects uncertainty in the appropriate average value to take for the different geometrical quantities due to topographic variation in the outlet. Are those two things the same?*

This statement was indeed misleading and we deleted it.

*On its own this explanation of running a separate set of experiments with reduced C is a bit confusing. I would make it clear that the first set of simulations had a non-negligible C, but in both cases the scaling relation held, which suggests the conclusions in this paper are unaffected by ignoring the basal friction in the derivation.*

We thank the Referee for pointing this out and removed this confusing explanation of the experiments. These experiments are already introduced in Sect. 3.2, which we now modified (P7, L195-200), also mentioning the deviations between the simulations and theory.

*Should this be 0.07 not 0.7?*

Yes! Corrected.